# Learning to Actively Learn: A Robust Approach

## Abstract

This work proposes a procedure for designing algorithms for specific adaptive data collection tasks like active learning and pure-exploration multi-armed bandits. Unlike the design of traditional adaptive algorithms that rely on concentration of measure and careful analysis to justify the correctness and sample complexity of the procedure, our adaptive algorithm is learned via adversarial training over equivalence classes of problems derived from information theoretic lower bounds. In particular, a single adaptive learning algorithm is learned that competes with the best adaptive algorithm learned for each equivalence class. Our procedure takes as input just the available queries, set of hypotheses, loss function, and total query budget. This is in contrast to existing meta-learning work that learns an adaptive algorithm relative to an explicit, user-defined subset or prior distribution over problems which can be challenging to define and be mismatched to the instance encountered at test time. This work is particularly focused on the regime when the total query budget is very small, such as a few dozen, which is much smaller than those budgets typically considered by theoretically derived algorithms. We perform synthetic experiments to justify the stability and effectiveness of the training procedure, and then evaluate the method on tasks derived from real data including a noisy 20 Questions game and a joke recommendation task.

## 1 Introduction

Closed-loop learning algorithms use previous observations to inform what measurements to take next in a closed-loop in order to accomplish inference tasks far faster than any fixed measurement plan set in advance. For example, active learning algorithms for binary classification have been proposed that under favorable conditions require exponentially fewer labels than passive, random sampling to identify the optimal classifier (Hanneke et al., 2014). And in the multi-armed bandits literature, adaptive sampling techniques have demonstrated the ability to identify the "best arm" that optimizes some metric with far fewer experiments than a fixed design (Garivier & Kaufmann, 2016; Fiez et al., 2019). Unfortunately, such guarantees often either require simplifying assumptions that limit robustness and applicability, or appeal to concentration inequalities that are very loose unless the number of samples is very large (e.g., web-scale).

The aim of this work is a framework that achieves the best of both worlds: algorithms that learn through simulated experience to be as effective as possible with a tiny measurement budget (e.g., 20 queries), while remaining robust due to adversarial training. Our work fits into a recent trend sometimes referred to as *learning to actively learn* (Konyushkova et al., 2017; Bachman et al., 2017; Fang et al., 2017; Boutilier et al., 2020; Kveton et al., 2020) which tunes existing algorithms or learns entirely new active learning algorithms by policy optimization. Previous works in this area learn a policy by optimizing with respect to data observed through prior experience (e.g., meta-learning or transfer learning) or an assumed explicit prior distribution of problem parameters (e.g. the true weight vector for linear regression). In contrast, our approach makes no assumptions about what parameters are likely to be encountered at test time, and therefore produces algorithms that do not suffer from a potential mismatch of priors. Instead, our method learns a policy that attempts to mirror the guarantees of frequentist algorithms with *instance dependent sample complexities*: if the problem is hard you will suffer a large loss, if it is easy you will suffer little.

The learning framework is general enough to be applied to many active learning settings of interest and is intended to be used to produce novel and robust high performing algorithms. The difference is that instead of hand-crafting hard instances that witness the difficulty of the problem, we use adversarial training inspired by the robust reinforcement learning literature to automatically train minimax policies. Embracing the use of a simulator allows our learned policies to be very aggressive while maintaining robustness. Indeed, this work is particularly useful in the setting where relatively few rounds of querying can be made, where concentration inequalities of existing algorithms are vacuous. To demonstrate the efficacy of our approach we implement the framework for the (transductive) linear bandit problem. This paradigm includes pure-exploration combinatorial bandits (e.g., shortest path, matchings) as a special case which itself reduces to active binary classification. We empirically validate our framework on a simple synthetic experiment before turning our attention to datasets derived from real data including a noisy 20 questions game and a joke recommendation task.

## 2    PROPOSED FRAMEWORK FOR ROBUST LEARNING TO ACTIVELY LEARN

Whether learned or defined by an expert, any algorithm for active learning can be thought of as a policy from the perspective of reinforcement learning. At time $t$, based on an internal state $s_t$, the policy takes action $x_t$ and receives observation $y_t$, which then updates the state and the process repeats. In our work, at time $t$ the state $s_t \in \mathcal{S}$ is a function of the history $\{(x_i, y_i)\}_{i=1}^{t-1}$ such as its sufficient statistics. Without loss of generality, a policy $\pi$ takes a state as input and defines a probability distribution over $\mathcal{X}$ so that at time $t$ we have $x_t \sim \pi(s_t)$. Fix a horizon $T$. For $t = 1, 2, \ldots, T$

- state $s_t \in \mathcal{S}$ is a function of the history, $\{(x_i, y_i)\}_{i=1}^{t-1}$,
- action $x_t \in \mathcal{X}$ is drawn at random from the distribution $\pi(s_t)$ defined over $\mathcal{X}$, and
- next state $s_{t+1} \in \mathcal{S}$ is constructed by taking action $x_t$ in state $s_t$ and observing $y_t \sim f(\cdot|\theta_*, s_t, x_t)$

until the game terminates at time $t = T$ and the policy receives loss $L_T$. Note that $L_T$ is a random variable that depends on the tuple $(\pi, \{(x_i, y_i)\}_{i=1}^{T}, \theta_*)$. We assume that $f$ is a distribution of known parameteric form to the policy (e.g., $f(\cdot|\theta, s, x) \equiv \mathcal{N}(\langle x, \theta \rangle, 1)$) but the parameter $\theta$ is unknown to the policy. Let $\mathbb{P}_{\pi,\theta}, \mathbb{E}_{\pi,\theta}$ denote the probability and expectation under the probability law induced by executing policy $\pi$ in the game with $\theta_* = \theta$ to completion. Note that $\mathbb{P}_{\pi,\theta}$ includes any internal randomness of the policy $\pi$ and the random observations $y_t \sim f(\cdot|\theta, s_t, x_t)$. Thus, $\mathbb{P}_{\pi,\theta}$ assigns a probability to any trajectory $\{(x_i, y_i)\}_{i=1}^{T}$. For a given policy $\pi$ and $\theta_* = \theta$, the metric of interest we wish to minimize is the expected loss $\ell(\pi, \theta) := \mathbb{E}_{\pi,\theta}[L_T]$ where $L_T$ as defined above is the loss observed at the end of the episode. For a fixed policy $\pi$, $\ell(\pi, \theta)$ defines a loss surface over all possible values of $\theta$. This loss surface captures the fact that some values of $\theta$ are just intrinsically harder than others, but also that a policy may be better suited for some values of $\theta$ versus others.

*Example*: In active binary classification, $T$ is a label budget, $\mathcal{X}$ could be a set of images such that we can query the label of example image $x_t \in \mathcal{X}$, $y_t \in \{-1, 1\}$ is the requested binary label, and the loss $L_T$ is the classification error of a trained classifier on these collected labels. Finally, $\theta_x = p(y = 1|x)$ for all $x \in \mathcal{X}$. More examples can be found in Appendix A.

### 2.1    INSTANCE DEPENDENT PERFORMANCE METRIC

We now define the sense in which we wish to evaluate a particular policy. For any fixed value of $\theta$ one could clearly design an algorithm that would maximize performance on $\theta$, but then it might have very poor performance on some other value $\theta' \neq \theta$. Thus, we would ideally like $\pi$ to perform uniformly well over a set of $\theta$'s that are all equivalent in a certain sense. Define a positive function $\mathcal{C} : \Theta \to (0, \infty)$ that assigns a score to each $\theta \in \Theta$ that intuitively captures the "difficulty" of a particular $\theta$, and can be used as a partial ordering of $\Theta$. Ideally, $\mathcal{C}(\theta)$ is a monotonic transformation of $\ell(\widetilde{\pi}, \theta)$ for some "best" policy $\widetilde{\pi}$ that we will define shortly. We give the explicit $\mathcal{C}(\theta)$

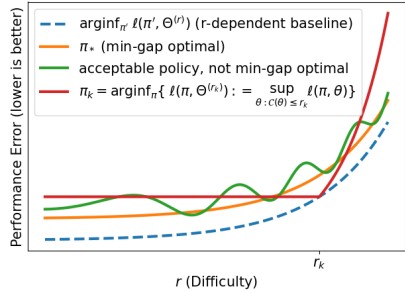

Figure 1: The $r$-dependent baseline defines a different policy for each value of $r$, thus, the blue curve may be unachievable with just a single policy. $\pi_*$ is the single policy that minimizes the maximum gap to this $r$-dependent baseline policy.

for the active binary classification example in Section 3, further description of $\mathcal{C}$ in Section 2.2, and more examples in Appendix A. For any set of problem instances $\Theta$ define

$$\ell(\pi, \Theta) := \sup_{\theta \in \Theta} \ell(\pi, \theta).$$

And for any $r \geq 0$, define

$$\Theta^{(r)} = \{\theta : \mathcal{C}(\theta) \leq r\}.$$

The quantity $\ell(\pi, \Theta^{(r)}) - \inf_{\pi'} \ell(\pi', \Theta^{(r)})$ is then a function of $r$ that describes the sub-optimality gap of a given policy $\pi$ relative to an $r$-dependent baseline policy trained specifically for each $r$. For a fixed $r_k > 0$, a policy $\pi$ that aims to minimize just $\ell(\pi, \Theta^{(r)})$ might focus just on the hard instances (i.e., those with $\mathcal{C}(\theta)$ close to $r$) and there may exist a different policy $\pi'$ that performs far better than $\pi$ on easier instances (i.e., those with $\mathcal{C}(\theta) \ll r$). To avoid this, assuming $\sup_r \ell(\pi, \Theta^{(r)}) - \inf_{\pi'} \ell(\pi', \Theta^{(r)}) < \infty$, we define

$$\pi_* := \arg\inf_{\pi} \sup_{r>0} \left( \ell(\pi, \Theta^{(r)}) - \inf_{\pi'} \ell(\pi', \Theta^{(r)}) \right) \tag{1}$$

as the policy that minimizes the worst case sub-optimality gap over all $r > 0$. Figure 1 illustrates these definitions. Instead of computing $\inf_{\pi'} \ell(\pi', \Theta^{(r)})$ for all $r$, in practice we define a grid with an increasing sequence $\{r_k\}_{k=1}^K$, to find an approximation to $\pi_*$. We are now ready to state the goal of this work:

---

**Objective:** Given an increasing sequence $r_1 < \cdots < r_K$ that indexes nested sets of problem instances of increasing difficulty, $\Theta^{(r_1)} \subset \Theta^{(r_2)} \subset \cdots \subset \Theta^{(r_K)}$, we wish to identify a policy $\widehat{\pi}$ that minimizes the maximum sub-optimality gap with respect to this sequence. Explicitly, we seek to learn

$$\widehat{\pi} := \arg\inf_{\pi} \max_{k \leq K} \left( \ell(\pi, \Theta^{(r_k)}) - \inf_{\pi'} \ell(\pi', \Theta^{(r_k)}) \right) \tag{2}$$

where $\ell(\pi, \Theta) := \sup_{\theta \in \Theta} \ell(\pi, \theta)$ and $\ell(\pi, \theta)$ is the expected loss incurred by policy $\pi$ on instance $\theta$.

---

Note that as $K \to \infty$ and $\sup_k \frac{r_{k+1}}{r_k} \to 1$, (1) and (2) are essentially equivalent under benign smoothness conditions on $\mathcal{C}(\theta)$, in which case $\widehat{\pi} \to \pi_*$. In practice, we choose a finite $K$ where $\Theta^{r_K}$ contains all problems that can be solved within the budget $T$ relatively accurately, and a small $\epsilon > 0$, where $\max_k \frac{r_{k+1}}{r_k} = 1 + \epsilon$. Furthermore, the objective in (2) is equivalent with

$$\widehat{\pi} = \arg\inf_{\pi} \max_{k \leq K} \left( \ell(\pi, \Theta^{(r_k)}) - \ell(\pi_k, \Theta^{(r_k)}) \right)$$

$$\text{where} \quad \pi_k \in \arg\inf_{\pi} \sup_{\theta : \mathcal{C}(\theta) \leq r_k} \ell(\pi, \theta).$$

We can efficiently solve this objective by first computing $\pi_k$ for all $k \in [K]$ to obtain $\ell(\pi_k, \Theta^{(r_k)})$ as benchmarks, and then use these benchmarks to train $\widehat{\pi}$.

## 2.2 PICKING THE COMPLEXITY FUNCTION $\mathcal{C}(\theta)$

We have defined an optimal policy in terms of a function $\mathcal{C}(\theta)$ that determines a partial ordering over instances $\theta$. This function can come from a heuristic that intuitively captures the difficulty of an instance. Or it can be defined and motivated from information theoretic lower bounds that often describe a general ordering, but are typically very loose relative to empirical performance. For example, consider the standard multi-armed bandit game where an agent has access to $K$ distributions and in each round $t \in [T]$ she chooses a distribution $I_t \in [K]$ and observes a random variable in $[0, 1]$ with mean $\theta_{I_t}$. If her strategy is described by a policy $\pi$, once $t$ reaches $T$ she receives loss $L_T = \max_{i \in [K]} \sum_{t=1}^T \theta_i - \theta_{I_t}$ with expectation $\ell(\pi, \theta) = \mathbb{E}[L_T]$ where the expectation is taken with respect to the randomness in the observations, and potentially any randomness of the policy. Under benign conditions, it is known that *any* policy must suffer $\ell(\pi, \theta) \gtrsim \min\{\sqrt{KT}, \sum_{i \neq *} (\theta_* - \theta_i)^{-1}\}$ where $\theta_* = \max_{i \in [k]} \theta_i$ (Lattimore & Szepesvári, 2018). Such a lower bound is an ideal candidate for $\mathcal{C}(\theta)$. We define a different $\mathcal{C}(\theta)$ for our particular experiments of interest, and others are described in Appendix A. The bottom line is that any function $\mathcal{C}(\theta)$ works, but if it happens to correspond to an information theoretic lower bound, the resulting policy will match the lower bound if it is achievable.

## 2.3 DIFFERENTIABLE POLICY OPTIMIZATION

The first step in learning the policy $\widehat{\pi}$ defined in Equation 2 is to learn each $\pi_k := \inf_\pi \sup_{\theta:\mathcal{C}(\theta)\leq r_k} \ell(\pi,\theta)$ for all $k = 1,\ldots,K$. Once all $\pi_k$ are defined, $\widehat{\pi}$ of (2) is an optimization of the same form after shifting the loss by the scalar $\ell(\pi_k,\Theta^{(r_k)})$. Consequently, to learn $\widehat{\pi}$ it suffices to develop a training procedure to solve $\inf_\pi \sup_{\theta\in\Omega} \ell'(\pi,\theta)$ for an arbitrary set $\Omega$ and generic loss function $\ell'(\pi,\theta)$.

To make the optimization problem $\inf_\pi \sup_{\theta\in\Omega} \ell'(\pi,\theta)$ tractable, we parameterize it as follows. First, to compute the suprema over $\Theta$, we consider a finite set $\widetilde{\Theta} := \{\widetilde{\theta}_i\}_{i=1}^N \subset \Omega$, weighted by SOFTMAX($w$) where $w \in \mathbb{R}^N$. In addition, instead of optimizing over all possible policies, we restrict the policy as the class of neural networks that take state representation as input and output a probability distribution over actions, parameterized by weights $\psi$. Mathematically, it could be stated as the following:

$$\inf_\pi \sup_{\theta\in\Omega} \ell(\pi,\theta) = \inf_\pi \sup_{\widetilde{\theta}_{1:N}\subset\Omega} \max_{i\in[N]} \ell(\pi,\widetilde{\theta}_i) \tag{3}$$

$$= \inf_\pi \sup_{w\in\mathbb{R}^N,\widetilde{\theta}_{1:N}\subset\Omega} \mathbb{E}_{i\sim\text{SOFTMAX}(w)}\left[\ell(\pi,\widetilde{\theta}_i)\right] \tag{4}$$

$$\approx \inf_\psi \sup_{w\in\mathbb{R}^N,\widetilde{\theta}_{1:N}\subset\Omega} \mathbb{E}_{i\sim\text{SOFTMAX}(w)}\left[\ell(\pi^\psi,\widetilde{\theta}_i)\right]. \tag{5}$$

---

**Algorithm 1:** Gradient Based Optimization of (5)

---

1 **Input**: partition $\Omega$, number of iterations $N_{it}$, number of problem samples $M$, number of rollouts per problem $L$, and loss variable $L_T$ at horizon $T$ (see beginning of Section 2).

2 **Goal**: Compute the optimal policy $\arg\inf_\pi \sup_{\theta\in\Omega} \ell(\pi,\theta) = \arg\inf_\pi \sup_{\theta\in\Omega} \mathbb{E}_{\pi,\theta}[L_T]$.

3 **Initialization**: $w$, finite set $\widetilde{\Theta}$ and $\psi$

4 **for** $t = 1,\ldots,N_{it}$ **do**

5     **Collect rollouts of play:**

6         Sample $M$ problem indices $I_1,\ldots,I_M \overset{i.i.d.}{\sim}$ SOFTMAX($w$)

7         **for** $m = 1,\ldots,M$ **do**

8             Collect $L$ independent rollout trajectories, denoted as $\tau_{m,1:L}$, by the policy $\pi^\psi$

            for problem instant $\theta_{I_m}$ and observe losses $\forall 1 \leq l \leq L, L_T(\pi^\psi,\tau_{m,l},\widetilde{\theta}_{I_m})$.

9         **end**

10     **Optimize worst cases in $\Omega$:**

11         Update the generating distribution by taking ascending steps on gradient estimates:

$$w \leftarrow w + \frac{1}{ML}\sum_{m=1}^M \nabla_w \log(\text{SOFTMAX}(w)_{I_m}) \cdot (\sum_{l=1}^L L_T(\pi^\psi,\tau_{m,l},\widetilde{\theta}_{I_m})) \tag{6}$$

$$\widetilde{\Theta} \leftarrow \widetilde{\Theta} + \frac{1}{ML}\sum_{m=1}^M \sum_{l=1}^L \left(\nabla_{\widetilde{\Theta}}\mathcal{L}_{\text{barrier}}(\widetilde{\theta}_{I_m},\Omega) + \nabla_{\widetilde{\Theta}} L_T(\pi^\psi,\tau_{m,l},\widetilde{\theta}_{I_m})\right.$$
$$\left. + L_T(\pi^\psi,\tau_{m,l},\widetilde{\theta}_{I_m}) \cdot \nabla_{\widetilde{\Theta}} \log(\mathbb{P}_{\pi^\psi,\widetilde{\theta}_{I_m}}(\tau_{m,l}))\right) \tag{7}$$

        where $\mathcal{L}_{\text{barrier}}$ is a differentiable barrier loss that heavily penalizes the $\widetilde{\theta}_{I_m}$'s outside $\Omega$.

12     **Optimize policy:**

13         Update the policy by taking descending step on gradient estimate:

$$\psi \leftarrow \psi - \frac{1}{ML}\sum_{m=1}^M \sum_{l=1}^L L_T(\pi^\psi,\tau_{m,l},\widetilde{\theta}_{I_m}) \cdot \nabla_\psi \log(\mathbb{P}_{\pi^\psi,\widetilde{\theta}_{I_m}}(\tau_{m,l})) \tag{8}$$

14 **end**

---

Note that the objectives in (3) and (4) are indeed equivalent as $\widetilde{\theta}_{1:N}$ are free parameters we optimize over rather than taking fixed values. Now, to motivate (5), starting from the left hand side of (3),

observe that a small change in $\pi$ may result in a large change in $\mathrm{argsup}_{\theta \in \Omega} \ell(\pi, \theta)$. Therefore, with the goal of covering the entire $\Omega$, we optimize the $N$ points so that when $\pi$ changes a bit, there is at least one $\widetilde{\theta}_i$ close to the optimal argsup. In addition, to covering the entire space of $\Omega$, $N$ is expected to be very large in practice. However, to optimize the objective effectively, we can only evaluate on $M$ of $\widetilde{\theta}$'s ($M \ll N$) in each iteration. Therefore, instead of naively sampling $M$ points uniformly at random from the $N$ points, in (4), we optimize an extra multinomial distribution, SOFTMAX($w$), over the $N$ points so that the points around the argsup are sampled more often. The final approximation in (5) comes from parameterizing the policy by a neural network.

To solve the saddle point optimization problem in (5), we use an instance of the Gradient Descent Ascent (GDA) algorithm as shown in Algorithm 1. The gradient estimates are unbiased estimates of the true gradients with respect to $\psi$, $w$ and $\widetilde{\Theta}$ (shown in Appendix B). We choose $N$ large enough to avoid *mode collapse*, and $M, L$ as large as possible to reduce variance in gradient estimates while fitting the memory constraint. We use Adam optimization (Kingma & Ba, 2014) in taking gradient updates and regularize some of the parameters (an example will be presented in the next section).

Note the decomposition for $\log(\mathbb{P}_{\pi^\psi, \theta'}(\tau))$ in (7) and (8), where rollout $\tau = \{(x_t, y_t)\}_{t=1}^T$, and

$$\log(\mathbb{P}_{\pi^\psi, \theta'}(\{(x_t, y_t)\}_{t=1}^T)) = \log\left(\pi^\psi(x_1) \cdot f(y_1|\theta', s_1) \cdot \prod_{t=2}^T \pi^\psi(s_t, x_t) \cdot f(y_t|\theta', s_t, x_t)\right).$$

Here $\pi^\psi$ and $f$ are only dependent on $\psi$ and $\widetilde{\Theta}$ respectively. During evaluation of a fixed policy $\pi$, we are interested in solving $\sup_{\theta \in \Omega} \ell(\pi, \theta)$ by gradient ascent updates like (7). The decoupling of $\pi^\psi$ and $f$ thus enables us to optimize the objective without differentiating through a policy $\pi$, which could be non-differentiable policies like deterministic algorithms.

Finally, we make a few remarks on the parameterization of (5). As given in (5), we represent the generating distribution $\mathcal{P}$ as a simple finite number of weighted particles, analogous to a particle filter. Our policy parameterization $\pi^\psi$ could be modelled by multi-layer perceptrons, recurrent neural networks, etc. We note that when using alternative generator parameterization like GANs (Goodfellow et al., 2014), an unbiased gradient can also be derived similarly.

## 3 IMPLEMENTATION FOR LINEAR BANDITS AND CLASSIFICATION

We now apply the general framework of the previous section to a specific problem: transductive linear bandits. As described in Sections 5 and Appendix A this setting generalizes standard multi-armed bandits, linear bandits, and all of binary classification through a simple reduction to combinatorial bandits. We are particularly motivated to look at classification because the existing agnostic active learning algorithms are very inefficient (see Section 5). Indeed when applied to our setting of $T = 20$ they never get past their first stage of uniform random sampling. Consider the game:

---

**Input**: Policy $\pi$, $\mathcal{X} \subset \mathbb{R}^d$, $\mathcal{Z} \subset \mathbb{R}^d$, time horizon $T \in \mathbb{T}$
**Initialization**: Nature chooses $\theta_* \in \mathbb{R}^d$ (hidden from policy)
**for** $t = 1, 2, \ldots, T$
- Policy $\pi$ selects $x_t \in \mathcal{X}$ using history $\{(x_s, y_s)\}_{s=1}^{t-1}$
- Nature reveals $y_t \sim f(\cdot|\theta_*, x_t)$ with $\mathbb{E}[y_t|\theta_*, x_t] = \langle x_t, \theta_* \rangle$

**Output**: Policy $\pi$ recommends $\widehat{z} \in \mathcal{Z}$ as an estimate for $z^\star(\theta_*) := \mathrm{argmax}_{z \in \mathcal{Z}} \langle z, \theta_* \rangle$ and

suffers loss $L_T = \begin{cases} \langle z^\star(\theta_*) - \widehat{z}, \theta_* \rangle & \text{if SIMPLE REGRET} \\ \mathbf{1}\{z^\star(\theta_*) \neq \widehat{z}\} & \text{if BEST IDENTIFICATION} \end{cases}$

---

The observation distribution $f(\cdot|\theta, x)$ is domain specific but typically taken to be either a Bernoulli distribution for binary data, or Gaussian for real-valued data. We are generally interested in two objectives: BEST IDENTIFICATION which attempts to exactly identify the vector $z^\star \in \mathcal{Z}$ that is most aligned with $\theta_*$, and SIMPLE REGRET which settles for an approximate maximizer.

**Defining** $\mathcal{C}(\theta)$ Recalling the discussion of Section 2.1, $\mathcal{C}(\theta)$ should ideally be monotonically increasing in the intrinsic difficulty of minimizing the loss with respect to a particular $\theta$. For arbitrary $\mathcal{X} \subset \mathbb{R}^d$ and $\mathcal{Z} \subset \mathbb{R}^d$, it is shown in Fiez et al. (2019) that the sample complexity of identifying $z^\star(\theta) = \arg\max_{z \in \mathcal{Z}} \langle z, \theta \rangle$ with high probability is proportional to a quantity $\rho_\star(\theta)$, the value

obtained by an optimization program. Another complexity term that appears in the combinatorial bandits literature (Cao & Krishnamurthy, 2017) where $\mathcal{X} = \{\mathbf{e}_i : i \in [d]\}$ and $\mathcal{Z} \subset \{0, 1\}^d$ is

$$\widetilde{\rho}(\theta) = \sum_{i=1}^{d} \max_{z:z_i \neq z_i^{\star}(\theta)} \frac{\|z - z^{\star}(\theta)\|_2^2}{\langle z - z^{\star}(\theta), \theta \rangle^2}. \tag{9}$$

One can show $\rho^{\star}(\theta) \leq \widetilde{\rho}(\theta)$ and in many cases track each other. Because $\widetilde{\rho}(\theta)$ can be computed much more efficiently compared to $\rho_{\star}(\theta)$, we use $\mathcal{C}(\theta) = \widetilde{\rho}(\theta)$ in our experiments.

---

**Algorithm 2:** Training Workflow

---

1 **Input**: sequence $\{r_k\}_{k=1}^{K}$, complexity function $\mathcal{C}$, and $obj \in \{\text{SIMPLE REGRET, BEST IDENTIFICATION}\}$.
2 **Define** $k(\theta) \in [K]$ such that $r_{k(\theta)-1} < \mathcal{C}(\theta) \leq r_{k(\theta)}$ for all $\theta$ with $\mathcal{C}(\theta) \leq r_K$
3 For each $k \in [K]$, obtain policy $\widetilde{\pi}_k$ by Algorithm 1 with $\Omega = \Theta^{(r_k)}$ and SIMPLE REGRET loss
4 **if** $obj$ is SIMPLE REGRET **then**
5      For each $k \in [K]$, compute $\ell(\widetilde{\pi}_k, r_k)$      // In this case, $\pi_k = \widetilde{\pi}_k$.
6      Warm start $\widehat{\pi} = \widetilde{\pi}_{\lfloor K/2 \rfloor}$; optimize $\widehat{\pi}$ by Algorithm 1 with $\Omega = \Theta^{(r_K)}$ and objective in (2),      i.e., $L_T = \langle z^{\star}(\theta) - \widehat{z}, \theta \rangle - \ell(\widetilde{\pi}_{k(\theta)}, \Theta^{(r_{k(\theta)})})$
7 **else if** $obj$ is BEST IDENTIFICATION **then**
8      For each $k \in [K]$, warm start $\pi_k = \widetilde{\pi}_k$; optimize $\pi_k$ by Algorithm 1 with $\Omega = \Theta^{(r_k)}$ and      BEST IDENTIFICATION loss; compute $\ell(\pi_k, \Theta^{(r_k)})$
9      Warm start $\widetilde{\pi} = \pi_{\lfloor K/2 \rfloor}$; optimize $\widetilde{\pi}$ by Algorithm 1 with $\Omega = \Theta^{(r_K)}$ and objective in (2),      i.e., $L_T = \mathbf{1}\{z^{\star}(\theta) \neq \widehat{z}\} - \ell(\pi_{k(\theta)}, \Theta^{(r_{k(\theta)})})$
10 **end**
11 **Output**: $\widehat{\pi}$ (an approximate solution to (2))

---

**Training.** When training our policies, we follow the following procedure in Algorithm 2. Note that even when we are training for BEST IDENTIFICATION, we still warm start the training with optimizing SIMPLE REGRET. This is because a random initialized policy performs so poorly that BEST IDENTIFICATION is nearly always 1, making it difficult to improve the policy. In addition, our generating distribution parameterizations exactly follows from Section 2.3, while detailed state representations, policy parametrization and hyperparamters can be found in Appendix C, D.

**Loss functions.** Instead of optimizing the approximated quantity from (5) directly, we add regularizers to the losses for both the policy and generator. First, we choose the $\mathcal{L}_{\text{barrier}}$ in (7) to be $\lambda_{\text{barrier}} \cdot \max\{0, \log(\mathcal{C}(\mathcal{X}, \mathcal{Z}, \theta)) - \log(r_k)\}$, for some large constant $\lambda_{\text{barrier}}$. To discourage the policy from over committing to a certain action and/or the generating distribution from covering only a small subset of particles (i.e., mode collapse), we also add negative entropy penalties to both policy's output distributions and SOFTMAX($w$) with scaling factors $\lambda_{\text{Pol-reg}}$ and $\lambda_{\text{Gen-reg}}$.

## 4 EXPERIMENTS

We now evaluate the approach described in the previous section for combinatorial bandits with $\mathcal{X} = \{\mathbf{e}_i : i \in [d]\}$ and $\mathcal{Z} \subset \{0, 1\}^d$. We stress that the framework implemented here can be applied to *any* $\mathcal{X}, \mathcal{Z} \subset \mathbb{R}^d$ and any appropriate $f$–just plug and play to learn a new policy. In our experiments we take particular instances of combinatorial bandits with Bernoulli observations. We evaluated based on two criterion: instance-dependent worst-case and average-case. For instance-dependent worst-case, we measure, for each $r_k$ and policy $\pi$, $\ell(\pi, \Theta^{(r_k)}) := \max_{\theta \in \Theta^{(r_k)}} \ell(\pi, \theta)$ and plot this value as a function of $r_k$. We note that our algorithm is designed to optimize for such metric. For the secondary average-case metric, we instead measure, for policy $\pi$ and some collected set $\Theta$, $\frac{1}{|\Theta|} \sum_{\theta \in \Theta} \ell(\pi, \theta)$. Performances of instance-dependent worst-case metric are reported in Figures 2, 3, 4, 6, and 7 below while the average case performances are reported in the tables and Figure 5. Full scale of the figures can also be found in Appendix F.

**Algorithms.** We compare against a number of baseline active learning algorithms (see Section 5 for a review). UNCERTAINTY SAMPLING at time $t$ computes the empirical maximizer of $\langle z, \widehat{\theta} \rangle$ and the runner-up, and samples an index uniformly from their symmetric difference; if either are not unique, an index is sampled from the region of disagreement of the winners (see Appendix G for details). The greedy methods are represented by soft generalized binary search (SGBS) (Nowak, 2011) which maintains a posterior distribution over $\mathcal{Z}$ and samples to maximize information gain. A hyperparameter $\beta \in (0, 1/2)$ of SGBS determines the strength of the likelihood update. We plot or report a range of performance over $\beta \in \{.01, .03, .1, .2, .3, .4\}$. The agnostic algorithms for classification (Dasgupta, 2006; Hanneke, 2007b;a; Dasgupta et al., 2008; Huang et al., 2015; Jain & Jamieson, 2019) or combinatorial bandits (Chen et al., 2014; Gabillon et al., 2016; Chen et al., 2017; Cao & Krishnamurthy, 2017; Fiez et al., 2019; Jain & Jamieson, 2019) are so conservative that given just $T = 20$ samples, they are all exactly equivalent to uniform sampling and hence represented by UNIFORM. To represent a policy based on learning to actively learn (LAL), we employ the method of Kveton et al. (2020) with a fixed prior $\widetilde{\mathcal{P}}$ constructed by drawing a $z$ uniformly at random from $\mathcal{Z}$ and defining $\theta = 2z - 1 \in [-1, 1]^d$ (details in Appendix H). When evaluating each policy, we use the successive halving algorithm (Li et al., 2017; 2018) for optimizing our non-convex objective with randomly initialized gradient descent and restarts (details in Appendix E).

**Thresholds.** We begin with a very simple instance to demonstrate the instance-dependent performance achieved by our learned policy. For $d = 25$ let $\mathcal{X} = \{\mathbf{e}_i : i \in [d]\}$, $\mathcal{Z} = \{\mathbf{0} + \sum_{i=1}^{k} \mathbf{e}_i : k = 0, 1, \ldots, d\}$, and $f(\cdot|\theta, x)$ is a Bernoulli distribution over $\{-1, 1\}$ with mean $\langle x, \theta \rangle \in [-1, 1]$. Note this is a binary classification task in one-dimension where the set of classifiers are thresholds on a line. We trained baseline policies $\{\pi_k\}_{k=1}^9$ for the BEST IDENTIFICATION metric with $\mathcal{C}(\theta) = \widetilde{\rho}(\mathcal{X}, \mathcal{Z}, \theta)$ and $r_k = 2^{3+i/2}$ for $i \in \{0, \ldots, 8\}$.

First we compare the base policies $\pi_k$ to $\widehat{\pi}$. Figure 2 presents $\ell(\pi, \Theta^{(r)}) = \sup_{\theta:\widetilde{\rho}(\theta) \leq r} \ell(\pi, \theta) = \sup_{\theta:\widetilde{\rho}(\theta) \leq r} \mathbb{P}_{\pi,\theta}(\widehat{z} \neq z_\star(\theta))$ as a function of $r$ for our base policies $\{\pi_k\}_k$ and the global policy $\pi_\star$, each as an individual curve. Figure 3 plots the same information in terms of gap: $\ell(\pi, \Theta^{(r)}) - \min_{k:r_{k-1} < r \leq r_k} \ell(\pi_k, \Theta^{(r_k)})$. We observe that each

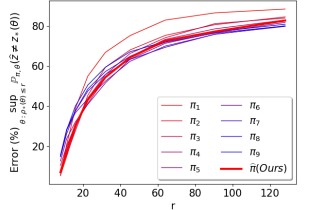

Figure 2: Learned policies, lower is better

Figure 3: Sub-optimality of individual policies, lower is better

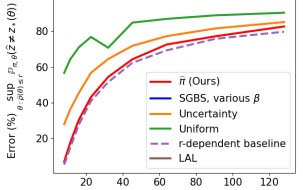

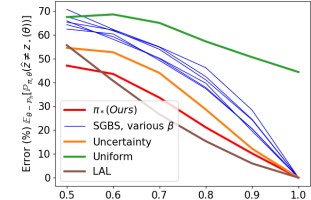

Figure 4: Max $\{\theta : \widetilde{\rho}(\theta) \leq r\}$, lower is better

Figure 5: Average $\mathbb{E}_{\theta \sim \mathcal{P}_h}[\cdot]$, lower is better

$\pi_k$ performs best in a particular region and $\pi_\star$ performs almost as well as the $r$-dependent baseline policies over the range of $r$. *This plot confirms that our optimization objective of* (2) *was successful.* Under the same conditions as Figure 2, Figure 4 compares the performance of $\pi_\star$ to the algorithm benchmarks. Since SGBS and LAL are deterministic, the adversarial training finds a $\theta$ that tricks them into catastrophic failure. Figure 5 trades adversarial evaluation for evaluating with respect to a parameterized prior: For each $h \in \{0.5, 0.6, \ldots, 1\}$, $\theta \sim \mathcal{P}_h$ is defined by drawing a $z$ uniformly at random from $\mathcal{Z}$ and then setting $\theta_i = (2z_i - 1)(2\alpha_i - 1)$ where $\alpha_i \sim \text{Bernoulli}(h)$. Thus, each sign of $2z-1$ is flipped with probability $h$. We then compute $\mathbb{E}_{\theta \sim \mathcal{P}_h}[\mathbb{P}_{\pi,\theta}(\widehat{z} = z_\star(\theta))] = \mathbb{E}_{\theta \sim \mathcal{P}_h}[\ell(\pi, \theta)]$. While SGBS now performs much better than uniform and uncertainty sampling, our policy $\pi_\star$ is still superior to these policies. However, LAL is best overall which is expected since the support of $\mathcal{P}_h$ is basically a rescaled version of the prior used in LAL.

**20 Questions.** We now address an instance constructed from the real data of Hu et al. (2018). Summarizing how we used the data from Hu et al. (2018) (see Appendix I for details), 100 yes/no questions were considered for 1000 celebrities. Each question $i \in [100]$ for each person $j \in [1000]$ was answered by several annotators to construct an empirical probability $\bar{p}_i^{(j)} \in [0, 1]$ denoting the

proportion of annotators that answered "yes." To construct our instance, we take $\mathcal{X} = \{\mathbf{e}_i : i \in [100]\}$ and $\mathcal{Z} = \{z^{(j)} : [z_i^{(j)}] = \mathbf{1}\{\bar{p}_i^{(j)} > 1/2\}\} \subset \{0,1\}^{1000}$. Just as before, we trained $\{\pi_k\}_{k=1}^4$ for the BEST IDENTIFICATION metric with $\mathcal{C}(\theta) = \widetilde{\rho}(\mathcal{X}, \mathcal{Z}, \theta)$ and $r_i = 2^{3+i/2}$ for $i \in \{1, \ldots, 4\}$.

Figure 6 is analogous to Figure 4 but for this 20 questions instance. Uncertainty sampling performs remarkably well on this instance. A potential explanation is that on a noiseless instance (e.g., $\theta = 2z - 1$ for some $z \in \mathcal{Z}$), our implementation of uncertainty sampling is equivalent to CAL (Cohn et al., 1994) and is

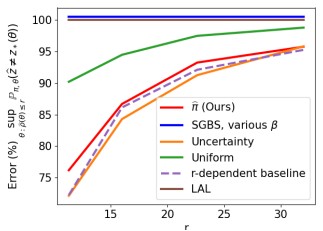

Figure 6: Max $\{\theta : \widetilde{\rho}(\theta) \leq r\}$

Table 1: Average $\mathbb{E}_{\theta \sim \widehat{\mathcal{P}}}[\cdot]$

| Method | Accuracy (%) |
|---|---|
| $\pi^*$ (Ours) | 17.9 |
| SGBS | {26.5, 26.2, 27.2, 26.5, 21.4, 12.8} |
| Uncertainty | 14.3 |
| LAL | 4.1 |
| Uniform | 6.9 |

known to have near-optimal sample complexity (Hanneke et al., 2014). Uncertainty sampling even outperforms our $r$-dependent baseline by a bit which in theory should not occur–we conjecture this is due to insufficient convergence of our policies or local minima. Our second experiment constructs a distribution $\widehat{\mathcal{P}}$ based on the dataset: to draw a $\theta \sim \widehat{\mathcal{P}}$ we uniformly at random select a $j \in [1000]$ and sets $\theta_i = 2\bar{p}_i^{(j)} - 1$ for all $i \in [d]$. As shown in Table 1, SGBS and $\pi_*$ are the winners. LAL performs much worse in this case, potentially because of the distribution shift from $\widetilde{\mathcal{P}}$ (prior we train on) to $\widehat{\mathcal{P}}$ (prior at test time). The strong performance of SGBS may be due to the fact that $\mathrm{sign}(\theta_i) = 2z_\star(\theta)_i - 1$ for all $i$ and $\theta \sim \widehat{\mathcal{P}}$, a realizability condition under which SGBS has strong guarantees (Nowak, 2011).

**Jester Joke Recommendation**
We now turn our attention away from best identification of the last two experiments $\ell(\pi, \theta) = \mathbb{P}_{\pi,\theta}(\widehat{z} \neq z_\star(\theta))$, to simple regret $\ell(\pi, \theta) = \mathbb{E}_{\pi,\theta}[\langle z_\star(\theta) - \widehat{z}, \theta\rangle]$. We consider the Jester jokes dataset of Goldberg et al. (2001) that contains jokes ranging innocent puns to grossly of-

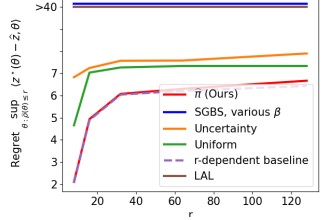

Figure 7: Max $\{\theta : \widetilde{\rho}(\theta) \leq r\}$

Table 2: Average $\mathbb{E}_{\theta \sim \widehat{\mathcal{P}}}[\cdot]$

| Method | Average Regret |
|---|---|
| $\pi^*$ (Ours) | 3.209 |
| SGBS | {3.180, 3.224, 3.278, 3.263, 3.153, 3.090} |
| Uncertainty | 3.027 |
| LAL | 3.610 |
| Uniform | 3.877 |

fensive jokes. We filter the dataset to only contain users that rated all 100 jokes, resulting in 14116 users. A rating of each joke was provided on a $[-10, 10]$ scale which was rescaled to $[-1, 1]$ and observations were simulated as Bernoulli's like above. We then clustered the ratings of these users (see Appendix J for details) to 10 groups to obtain $\mathcal{Z} = \{z^{(k)} : k \in [10], z^{(k)} \in \{0,1\}^{100}\}$ where $z_i^{(k)} = 1$ corresponds to recommending the $i$th joke in user cluster $z^{(k)} \in \mathcal{Z}$. Figure 7 shows the same style plot as Figures 4,6 but for this jokes dataset, with our policy alone nearly achieving the $r$-dependent baseline for all $r$. Mirroring the construction of the 20Q prior, we construct $\widehat{P}$ by uniformly sampling a user and employing their $\theta$ to answer queries. Table 2 shows that despite our policy not being trained for this setting, its performance is still among the top.

## 5 RELATED WORK

**Learning to actively learn.** Previous works vary in how the parameterize the policy, ranging from parameterized mixtures of existing expertly designed active learning algorithms (Baram et al., 2004; Hsu & Lin, 2015; Agarwal et al., 2016), parameterizing hyperparameters (e.g., learning rate, rate of forced exploration, etc.) in an existing popular algorithm (e.g, EXP3) (Konyushkova et al., 2017; Bachman et al., 2017; Cella et al., 2020), and the most ambitious, policies parameterized end-to-end like in this work (Boutilier et al., 2020; Kveton et al., 2020; Sharaf & Daumé III, 2019; Fang et al., 2017; Woodward & Finn, 2017). These works take an approach of defining a prior distribution either through past experience (meta-learning) or expert created (e.g., $\theta \sim \mathcal{N}(0, \Sigma)$), and then evaluate their policy with respect to this prior distribution. Defining this prior can be difficult, and moreover, if the $\theta$ encountered at test time did not follow this prior distribution, performance could suffer significantly. Our approach, on the other hand, takes an adversarial training approach and can

be interpreted as learning a parameterized least favorable prior (Wasserman, 2013), thus gaining a much more robust policy as an end result.

**Robust and Safe Reinforcement Learning:** Our work is also highly related to the field of robust and safe reinforcement learning, where our objective can be considered as an instance of *minimax criterion under parameter uncertainty* (Garcıa & Fernández, 2015). Widely applied in applications such as robotics (Mordatch et al., 2015; Rajeswaran et al., 2016), these methods train a policy in a simulator like Mujoco (Todorov et al., 2012) to minimize a defined loss objective while remaining robust to uncertainties and perturbations to the environment (Mordatch et al., 2015; Rajeswaran et al., 2016). Ranges of these uncertainty parameters are chosen based on potential values that could be encountered when deploying the robot in the real world. In our setting, however, defining the set of environments is far less straightforward and is overcome by the adoption of the $\mathcal{C}(\theta)$ function.

**Active Binary Classification Algorithms.** The literature on active learning algorithms can be partitioned into *model-based heuristics* like uncertainty sampling, query by committee, or model-change sampling (Settles, 2009), *greedy binary-search* like algorithms that typically rely on a form of bounded noise for correctness (Dasgupta, 2005; Kääriäinen, 2006; Golovin & Krause, 2011; Nowak, 2011), and *agnostic* algorithms that make no assumptions on the probabilistic model (Dasgupta, 2006; Hanneke, 2007b;a; Dasgupta et al., 2008; Huang et al., 2015; Jain & Jamieson, 2019). Though the heuristics and greedy methods can perform very well for some problems, it is typically easy to construct counter-examples (e.g., outside the assumptions) in which they catastrophically fail (as demonstrated in our experiments). The agnostic algorithms have strong robustness guarantees but rely on concentration inequalities, and consequently require at least hundreds of labels to observe any deviation from random sampling (see Huang et al. (2015) for comparison). Therefore, they were not included in our experiments explicitly but were represented by uniform.

**Pure-exploration Multi-armed Bandit Algorithms.** In the linear structure setting, for sets $\mathcal{X}, \mathcal{Z} \subset \mathbb{R}^d$ known to the player, pulling an "arm" $x \in \mathcal{X}$ results in an observation $\langle x, \theta_* \rangle +$ zero-mean noise, and the objective is to identify $\arg\max_{z \in \mathcal{Z}} \langle z, \theta_* \rangle$ for a vector $\theta_*$ unknown to the player (Soare et al., 2014; Karnin, 2016; Tao et al., 2018; Xu et al., 2017; Fiez et al., 2019). A special case of linear bandits is combinatorial bandits where $\mathcal{X} = \{\mathbf{e}_i : i \in [d]\}$ and $\mathcal{Z} \subset \{0, 1\}^d$ (Chen et al., 2014; Gabillon et al., 2016; Chen et al., 2017; Cao & Krishnamurthy, 2017; Fiez et al., 2019; Jain & Jamieson, 2019). Active binary classification is a special case of combinatorial pure-exploration multi-armed bandits (Jain & Jamieson, 2019), which we exploit in the threshold experiments. While the above works have made great theoretical advances in deriving algorithms and information theoretic lower bounds that match up to constants, the constants are so large that these algorithms only behave well when the number of measurements is very large. When applied to the instances of our paper (only 20 queries are made), these algorithms behave no differently than random sampling.

## 6    DISCUSSION AND FUTURE DIRECTIONS

We see this work as an exciting but preliminary step towards realizing the full potential of this general approach. From a practical perspective, training a $\pi_*$ can take many hours of computational resources for even these small instances. Scaling these methods to larger instances is an important next step. While training time scales linearly with the horizon length $T$, we note that one can take multiple samples per time step with minimal computational overhead enabling problems that require larger sample complexities. In our implementation we hard-coded the decision rule for $\hat{z}$ given $s_T$, while it could also be learned as in (Luedtke et al., 2020). Likewise, the parameterization of the policy and generator worked well for our purposes but was chosen somewhat arbitrarily–are there more natural choices? Finally, while we focused on stochastic settings, this work naturally extends to constrained fully adaptive adversarial sequences which is an interesting direction of future work.

FUNDING DISCLOSURE

Removed for anonymization purposes.

ACKNOWLEDGEMENT

Removed for anonymization purposes.

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

## A   INSTANCE DEPENDENT SAMPLE COMPLEXITY

Identifying forms of $\mathcal{C}(\theta)$ is not as difficult a task as one might think due to the proliferation of tools for proving lower bounds for active learning (Mannor & Tsitsiklis, 2004; Tsybakov, 2008; Garivier & Kaufmann, 2016; Carpentier & Locatelli, 2016; Simchowitz et al., 2017; Chen et al., 2014). One can directly extract values of $\mathcal{C}(\theta)$ from the literature for regret minimization of linear or other structured bandits (Lattimore & Szepesvari, 2016; Van Parys & Golrezaei, 2020), contextual bandits (Hao et al., 2019), and tabular as well as structured MDPs (Simchowitz & Jamieson, 2019; Ok et al., 2018). Moreover, we believe that even reasonable surrogates of $\mathcal{C}(\theta)$ should result in a high quality policy $\pi_*$.

We review some canonical examples:

- **Multi-armed bandits.** In the best-arm identification problem, there are $d \in \mathbb{N}$ Gaussian distributions where the $i$th distribution has mean $\theta_i \in \mathbb{R}$ for $i = 1, \ldots, d$. In the above formulation, this problem is encoded as action $x_t = i_t$ results in observation $y_t \sim \text{Bernoulli}(\theta_{i_t})$ and the loss $\ell(\pi, \theta) := \mathbb{E}_{\pi, \theta}[\mathbf{1}\{\widehat{i} \neq i_\star(\theta)\}]$ where $\widehat{i}$ is $\pi$'s recommended index and $i_\star(\theta) = \arg\max_i \theta_i$. It's been shown that there exists a constant $c_0 > 0$ such that for any sufficiently large $\nu > 0$ we have

$$\inf_\pi \sup_{\theta : \mathcal{C}_{MAB}(\theta) \leq \nu} \ell(\pi, \theta) \geq \exp(-c_0 T/\nu) \quad \text{where} \quad \mathcal{C}_{MAB}(\theta) := \sum_{i \neq i_\star(\theta)} (\theta_{i_\star(\theta)} - \theta_i)^{-2}$$

  Moreover, for any $\theta \in \mathbb{R}^d$ there exists a policy $\widetilde{\pi}$ that achieves $\ell(\widetilde{\pi}, \theta) \leq c_1 \exp(-c_2 T/\mathcal{C}_{MAB}(\theta))$ where $c_1, c_2$ capture constant and low-order terms (Carpentier & Locatelli, 2016; Karnin et al., 2013; Simchowitz et al., 2017; Garivier & Kaufmann, 2016).

The above correspondence between the lower bound and the upper bound suggests that $\mathcal{C}_{MAB}(\theta)$ plays a critical role in determining the difficult of identifying $i_\star(\theta)$ for *any* $\theta$. This exercise extends to more structured settings as well:

- **Content recommendation / active search.** Consider $n$ items (e.g., movies, proteins) where the $i$th item is represented by a feature vector $x_i \in \mathcal{X} \subset \mathbb{R}^d$ and a measurement $x_t = x_i$ (e.g., preference rating, binding affinity to a target) is modeled as a linear response model such that $y_t \sim \mathcal{N}(\langle x_i, \theta\rangle, 1)$ for some unknown $\theta \in \mathbb{R}^d$. If $\ell(\pi, \theta) := \mathbb{E}_{\pi, \theta}[\mathbf{1}\{\widehat{i} \neq i_\star(\theta)\}]$ as above then nearly identical results to that of above hold for an analogous function of $\mathcal{C}_{MAB}(\theta)$ (Soare et al., 2014; Karnin, 2016; Fiez et al., 2019).
- **Active binary classification.** For $i = 1, \ldots, d$ let $\phi_i \in \mathbb{R}^p$ be a feature vector of an unlabeled item (e.g., image) that can be queried for its binary label $y_i \in \{-1, 1\}$ where $y_i \sim \text{Bernoulli}(\theta_i)$ for some $\theta \in \mathbb{R}^d$. Let $\mathcal{H}$ be an arbitrary set of classifiers (e.g., neural nets, random forest, etc.) such that each $h \in \mathcal{H}$ assigns a label $\{-1, 1\}$ to each of the items $\{\phi_i\}_{i=1}^d$ in the pool. If items are chosen sequentially to observe their labels, the objective is to identify the true risk minimizer $h_\star(\theta) = \arg\min_{h \in \mathcal{H}} \sum_{i=1}^d \mathbb{E}_\theta[\mathbf{1}\{h(\phi_i) \neq y_i\}]$ using as few requested labels as possible and $\ell(\pi, \theta) := \mathbb{E}_{\pi, \theta}[\mathbf{1}\{\widehat{h} \neq h_\star(\theta)\}]$ where $\widehat{h} \in \mathcal{H}$ is $\pi$'s recommended classifier. Many candidates for $\mathcal{C}(\theta)$ have been proposed from the agnostic active learning literature (Dasgupta, 2006; Hanneke, 2007b;a; Dasgupta et al., 2008; Huang et al., 2015; Jain & Jamieson, 2019) but we believe the most granular candidates come from the combinatorial bandit literature (Chen et al., 2017; Fiez et al., 2019; Cao & Krishnamurthy, 2017; Jain & Jamieson, 2019). To make the reduction, for each $h \in \mathcal{H}$ assign a $z^{(h)} \in \{0, 1\}^d$ such that $[z^{(h)}]_i := \mathbf{1}\{h(\phi_i) = 1\}$ for all $i = 1, \ldots, d$ and set $\mathcal{Z} = \{z^{(h)} : h \in \mathcal{H}\}$. It is easy to check that $z_\star(\theta) := \arg\max_{z \in \mathcal{Z}}\langle z, \theta\rangle$ satisfies $z_\star(\theta) = z^{(h_\star(\theta))}$. Thus, requesting the label of example $i$ is equivalent to sampling from $\text{Bernoulli}(\langle \mathbf{e}_i, \theta\rangle) \in \{-1, 1\}$, completing the reduction to combinatorial bandits: $\mathcal{X} = \{\mathbf{e}_i : i \in [d]\}$, $\mathcal{Z} \subset \{0, 1\}^d$. We then apply the exact same $\mathcal{C}(\theta)$ as above for linear bandits.

## B   GRADIENT ESTIMATE DERIVATION

Here we derive the unbiased gradient estimates (6), (7) and (8) in Algorithm 1. Since each the gradient estimates in the above averages over $M \cdot L$ identically distributed trajectories, it is therefore sufficient to show that our gradient estimate is unbiased for a single problem $\widetilde{\theta}_i$ and its rollout trajectory $\{(x_t, y_t)\}_{t=1}^T$.

For a feasible $w$, using the score-function identity (Aleksandrov et al., 1968)

$$\nabla_w \mathbb{E}_{i \sim \text{SOFTMAX}(w)} \left[ \ell(\pi^\psi, \widetilde{\theta}_i) \right] = \mathbb{E}_{i \sim \text{SOFTMAX}(w)} \left[ \ell(\pi^\psi, \widetilde{\theta}_i) \cdot \nabla_w \log(\text{SOFTMAX}(w)_i) \right].$$

Observe that if $i \sim \text{SOFTMAX}(w)$ and $\{(x_t, y_t)\}_{t=1}^T$ is the result of rolling out a policy $\pi^\psi$ on $\widetilde{\theta}_i$ then

$$g^w := L_T(\pi^\psi, \{(x_t, y_t)\}_{t=1}^T, \widetilde{\theta}_i) \cdot \nabla_w \log(\text{SOFTMAX}(w)_i)$$

is an unbiased estimate of $\nabla_w \mathbb{E}_{i \sim \text{SOFTMAX}(w)} \left[ \ell(\pi^\psi, \widetilde{\theta}_i) \right]$.

For a feasible set $\widetilde{\Theta}$, by definition of $\ell(\pi, \theta)$,

$$\begin{aligned}
\nabla_{\widetilde{\Theta}} \mathbb{E}_{i \sim \text{SOFTMAX}(w)} \left[ \ell(\pi^\psi, \widetilde{\theta}_i) \right] &= \mathbb{E}_{i \sim \text{SOFTMAX}(w)} \left[ \nabla_{\widetilde{\Theta}} \mathbb{E}_{\pi, \widetilde{\theta}_i} \left[ L_T(\pi, \{(x_t, y_t)\}_{t=1}^T, \widetilde{\theta}_i) \right] \right] \\
&= \mathbb{E}_{i \sim \text{SOFTMAX}(w)} \left[ \mathbb{E}_{\pi, \widetilde{\theta}_i} \left[ \nabla_{\widetilde{\Theta}} L_T(\pi, \{(x_t, y_t)\}_{t=1}^T, \widetilde{\theta}_i) \right. \right. \qquad (10) \\
&\qquad \left. \left. + L_T(\pi, \{(x_t, y_t)\}_{t=1}^T, \widetilde{\theta}_i) \cdot \nabla_{\widetilde{\Theta}} \log(\mathbb{P}_{\pi^\psi, \widetilde{\theta}_i}(\{(x_t, y_t)\}_{t=1}^T)) \right] \right]
\end{aligned}$$

where the last equality follows from chain rule and the score-function identity (Aleksandrov et al., 1968). The quantity inside the expectations, call it $g^{\widetilde{\Theta}}$, is then an unbiased estimator of $\nabla_{\widetilde{\Theta}} \mathbb{E}_{i \sim \text{SOFTMAX}(w)} \left[ \ell(\pi^\psi, \widetilde{\theta}_i) \right]$ given $i$ and $\{(x_t, y_t)\}_{t=1}^T$ are rolled out accordingly. Note that if $\mathcal{L}_{\text{barrier}} \neq 0$, $\nabla_{\widetilde{\Theta}} \mathcal{L}_{\text{barrier}}(\widetilde{\theta}_i, \Omega)$ is clearly an unbiased gradient estimator of $\mathbb{E}_{i \sim \text{SOFTMAX}(w)} [\mathbb{E}_{\pi, \widetilde{\theta}_i} [\mathcal{L}_{\text{barrier}}(\widetilde{\theta}_i, \Omega)]]$ given $i$ and rollout are sampled accordingly.

Likewise, for policy,

$$g^\psi := L_T(\pi^\psi, \{(x_t, y_t)\}_{t=1}^T, \widetilde{\theta}_i) \cdot \nabla_\psi \log(\mathbb{P}_{\pi^\psi, \widetilde{\theta}_i}(\{(x_t, y_t)\}_{t=1}^T))$$

is an unbiased estimate of $\nabla_\psi \mathbb{E}_{i \sim \text{SOFTMAX}(w)} \left[ \ell(\pi^\psi, \widetilde{\theta}_i) \right]$.

## C  LINEAR BANDIT PARAMETERIZATION

### C.1  STATE REPRESENTATION

We parameterize our state space $\mathcal{S}$ as a flattened $|\mathcal{X}| \times 3$ matrix where each row represents a distinct $x \in \mathcal{X}$. Specifically, at time $t$ the row of $s_t$ corresponding to some $x \in \mathcal{X}$ records the number of times that action $x$ has been taken $\sum_{s=1}^{t-1} \mathbf{1}\{x_s = x\}$, its inverse $(\sum_{s=1}^{t-1} \mathbf{1}\{x_s = x\})^{-1}$, and the sum of the observations $\sum_{s=1}^{t-1} \mathbf{1}\{x_s = x\} y_s$.

### C.2  POLICY MLP ARCHITECTURE

Our policy $\pi^\psi$ is a multi-layer perceptron with weights $\psi$. The policy take a $3|\mathcal{X}|$ sized state as input and outputs a vector of size $|\mathcal{X}|$ which is then pushed through a soft-max to create a probability distribution over $\mathcal{X}$. At the end of the game, regardless of the policy's weights, we set $\widehat{z} = \text{argmax}_{z \in \mathcal{Z}} \langle z, \widehat{\theta} \rangle$ where $\widehat{\theta}$ is the minimum $\ell_2$ norm solution to $\text{argmin}_\theta \sum_{s=1}^T (y_s - \langle x_s, \theta \rangle)^2$.

Our policy network is a simple 6-layer MLP, with layer sizes $\{3|\mathcal{X}|, 256, 256, 256, 256, |\mathcal{X}|\}$ where $3|\mathcal{X}|$ corresponds to the input layer and $|\mathcal{X}|$ is the size of the output layer, which is then pushed through a Softmax function to create a probability over arms. In addition, all intermediate layers are activated with the leaky ReLU activation units with negative slopes of .01. For the experiments for 1D thresholds and 20 Questions, they share the same network structure as mentioned above with $|\mathcal{X}| = 25$ and $|\mathcal{X}| = 100$ respectively.

## D  HYPER-PARAMETERS

In this section, we list our hyperparameters. First we define $\lambda_{\text{binary}}$ to be a coefficient that gets multiplied to binary loses, so instead of $\mathbf{1}\{z^\star(\theta_*) \neq \widehat{z}\}$, we receive loss $\lambda_{\text{binary}} \cdot \mathbf{1}\{z^\star(\theta_*) \neq \widehat{z}\}$. We

choose $\lambda_{\text{binary}}$ so that the recieved rewards are approximately at the same scale as SIMPLE REGRET. During our experiments, all of the optimizers are Adam. All budget sizes are $T = 20$. For fairness of evaluation, during each experiment (1D thresholds or 20 Questions), all parameters below are shared for evaluating all of the policies. To elaborate on training strategy proposed in Algorithm 2 more, we divide our training into four procedures, as indicated in Table 3:

- **Init.** The initialization procedure takes up a rather small portion of iterations primarily for the purpose of optimizing for $\mathcal{L}_{\text{barrier}}$ so that the particles converge into the constrained difficulty sets. In addition, during the initialization process we initialize and freeze $w = \vec{0}$, thus putting an uniform distribution over the particles. This allows us to utilize the entire set of particles without $w$ converge to only a few particles early on. To initialize $\widetilde{\Theta}$, we sample 2/3 of the $N$ particles uniformly from $[-1, 1]^{|\mathcal{X}|}$ and the rest 1/3 of the particles by sampling, for each $i \in [|\mathcal{Z}|]$, $\frac{N}{3|\mathcal{Z}|}$ particles uniformly from $\{\theta : \text{argmax}_j \langle \theta, z_j \rangle = i\}$. We initialize our policy weights by Xavier initialization with weights sampled from normal distribution and scaled by .01.
- **Regret Training, $\widetilde{\pi}_i$** Training with SIMPLE REGRET objective usually takes the longest among the Procedures. The primary purpose for this process is to let the policy converge to a reasonable warm start that already captures some essence of the task.
- **Fine-tune $\pi_i$.** Training with BEST IDENTIFICATION objective run multiple times for each $\pi_i$ with their corresponding complexity set $\Theta_i$. During each run, we start with a warm started policy, and reinitialize the rest of the models by running the initialization procedure followed by optimizing the BEST IDENTIFICATION objective.
- **Fine-tune $\widehat{\pi}$** This procedure optimizes (2), with baselines $\min_k \ell(\pi_k, \Theta^{(r_k)})$ evaluated based on each $\pi_i$ learned from the previous procedure. Similar to fine-tuning each individual $\pi_i$, we warm start a policy $\pi_{\lfloor K/2 \rfloor}$ and reinitialize $w$ and $\Theta$ by running the initialization procedure again.

| | | Experiment | | |
|---|---|---|---|---|
| | | 1D Threshold | 20 Questions | Jester Joke |
| Procedure | Hyper-parameter | $\|\mathcal{X}\| = 25$ | $\|\mathcal{X}\| = 100$ | $\|\mathcal{X}\| = 100$ |
| Init | $N_{it}$ | | 20000 (all) | |
| | $\psi$ learning rate | | $10^{-4}$ (all) | |
| | $\widetilde{\Theta}$ learning rate | | $10^{-3}$ (all) | |
| | $w$ learning rate | | 0 (all) | |
| Regret Training | $N_{it}$ | | 480000 (all) | |
| | $\psi$ learning rate | | $10^{-4}$ (all) | |
| | $\widetilde{\Theta}$ learning rate | | $10^{-3}$ (all) | |
| | $w$ learning rate | | $10^{-3}$ (all) | |
| Fine-tune $\pi_i$ | $N_{it}$ | | 200000 (all) | |
| | $\psi$ learning rate | | $10^{-4}$ (all) | |
| | $\widetilde{\Theta}$ learning rate | | $10^{-3}$ (all) | |
| | $w$ learning rate | | $10^{-3}$ (all) | |
| Fine-tune $\pi_*$ | $N_{it}$ | 500000 | 300000 | 500000 |
| | $\psi$ learning rate | | $10^{-4}$ (all) | |
| | $\widetilde{\Theta}$ learning rate | | $10^{-3}$ (all) | |
| | $w$ learning rate | | $10^{-3}$ (all) | |
| Adam Optimizer | $\beta_1$ | | .9 (all) | |
| | $\beta_2$ | | .999 (all) | |

Table 3: Number of Iterations and Learning Rates

| | | Experiment | | |
|---|---|---|---|---|
| | | 1D Threshold | 20 Questions | Jester Joke |
| Procedure | Hyper-parameter | $\lvert \mathcal{X} \rvert = 25$ | $\lvert \mathcal{X} \rvert = 100$ | $\lvert \mathcal{X} \rvert = 100$ |
| | $N$ | $1000 \times \lvert \mathcal{Z} \rvert$ | $300 \times \lvert \mathcal{Z} \rvert$ | $2000 \times \lvert \mathcal{Z} \rvert$ |
| | M | 1000 | 500 | 500 |
| | L | 10 | 30 | 30 |
| Init + | $\lambda_{\text{binary}}$ | 7.5 | 30 | 30 |
| Train + | $\lambda_{\text{Pol-reg}}$(regret) | .2 | .8 | .8 |
| Fine-tune | $\lambda_{\text{Pol-reg}}$(fine-tune) | .3 | .8 | .8 |
| | $\lambda_{\text{Gen-reg}}$ | .05 | .1 | .05 |
| | $\lambda_{\text{barrier}}$ | $10^3$ (all) | | |

Table 4: Parallel Sizes and Regularization coefficients

To provide a general strategy of choosing hyper-parameters, we note that $L$, firstly, $\lambda_{\text{binary}}$, $\lambda_{\text{Pol-reg}}$ are primarily parameters tuned for $\lvert \mathcal{X} \rvert$ as the noisiness and scale of the gradients, and entropy over the arms $\mathcal{X}$ grows with the size $\lvert \mathcal{X} \rvert$. Secondly, $\lambda_{\text{Gen-reg}}$ is primarily tuned for $\lvert \mathcal{Z} \rvert$ as it penalizes the entropy over the $N$ arms, which is a multiple of $\lvert \mathcal{Z} \rvert$. Thirdly, learning rate of $\theta$ is primarily tuned for the convergence of constraint $\rho^*$ into the restricted class, thus $\mathcal{L}_{\text{barrier}}$ becoming 0 after the specified number of iterations during initialization is a good indicator. Finally, we choose $N$ and $M$ by memory constraint of our GPU. The hyper-parameters for each experiment was tuned with less than 20 hyper-parameter assignments, some metrics to look at while tuning these hyper-parameters includes but are not limited to: gradient magnitudes of each component, convergence of each loss and entropy losses for each regularization term (how close it is to the entropy of a uniform probability), etc.

## E  POLICY EVALUATION

When evaluating a policy, we are essentially solving the following objective for a fixed policy $\pi$:

$$\max_{\theta \in \Omega} \ell(\pi, \theta)$$

where $\Omega$ is a set of problems. However, due to non-concavity of this loss function, gradient descent initialized randomly may converge to a local maxima. To reduce this possibility, we randomly initialize many initial iterates and take gradient steps round-robin, eliminating poorly performing trajectories. To do this with a fixed amount of computational resource, we apply the successive halving algorithm from Li et al. (2018). Specifically, we choose hyperparamters: $\eta = 4$, $r = 100$, $R = 1600$ and $s = 0$. This translates to:

- Initialize $\lvert \widetilde{\Theta} \rvert = 1600$, optimize for 100 iterations for each $\widetilde{\theta}_i \in \widetilde{\Theta}$
- Take the top $400$ of them and optimize for another $400$ iterations
- Take the top $100$ of the remaining $400$ and optimize for an additional $1600$ iterations

We take gradient steps with the Adam optimizer (Kingma & Ba, 2014) with learning rate of $10^{-3}$ $\beta_1 = .9$ and $\beta_2 = .999$.

# F   FIGURES AT FULL SCALE

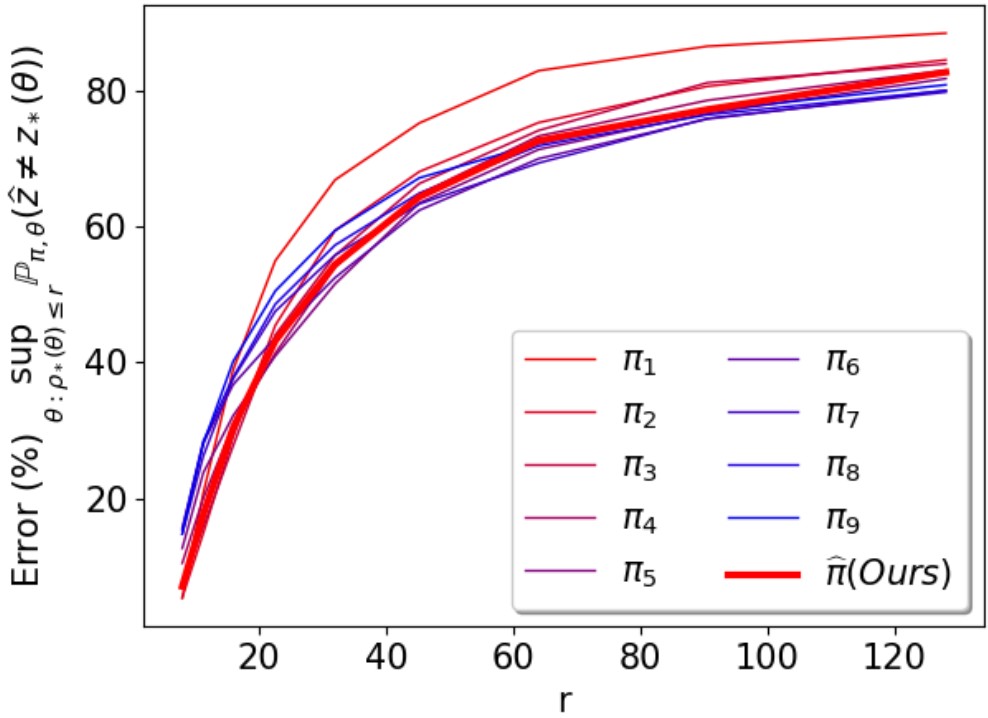

Figure 8: Full scale of Figure 2

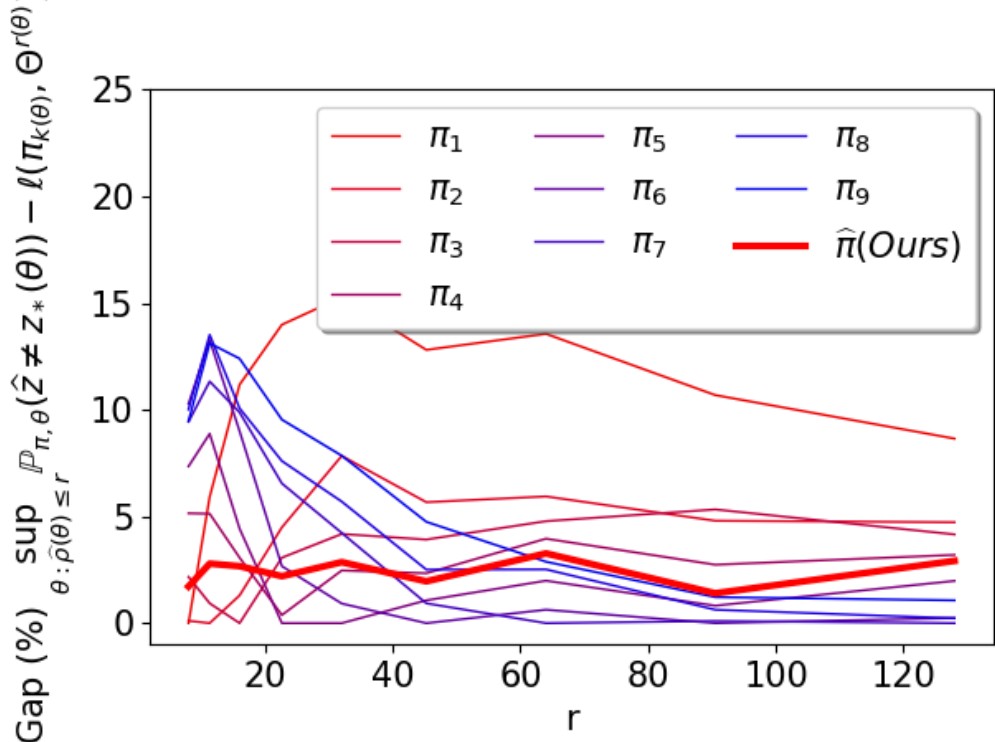

Figure 9: Full scale of Figure 3

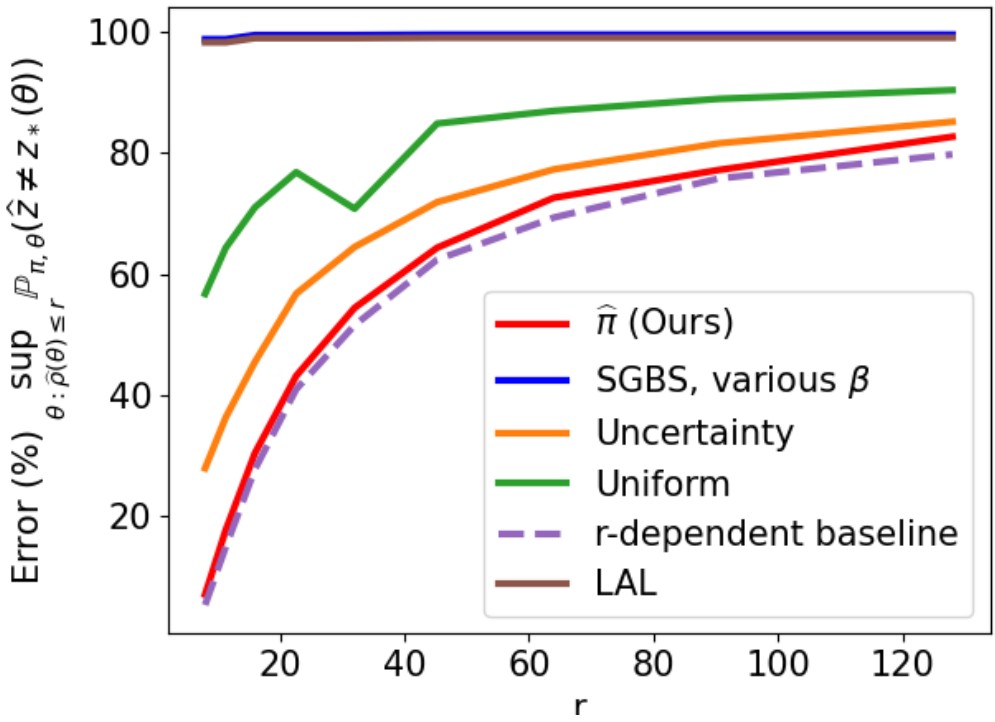

Figure 10: Full scale of Figure 4

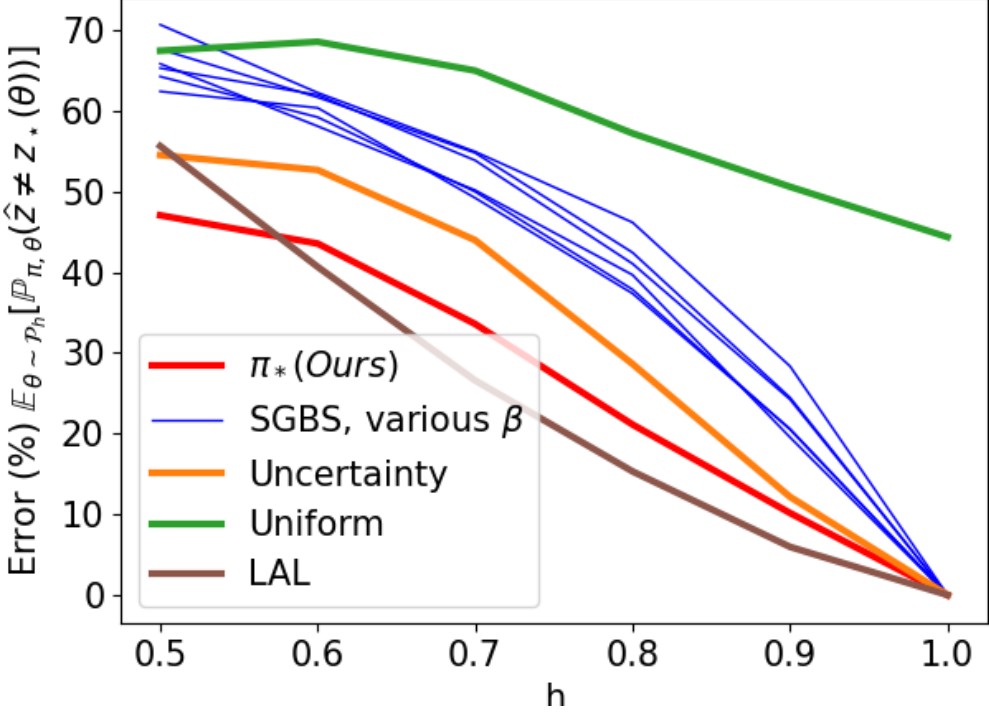

Figure 11: Full scale of Figure 5

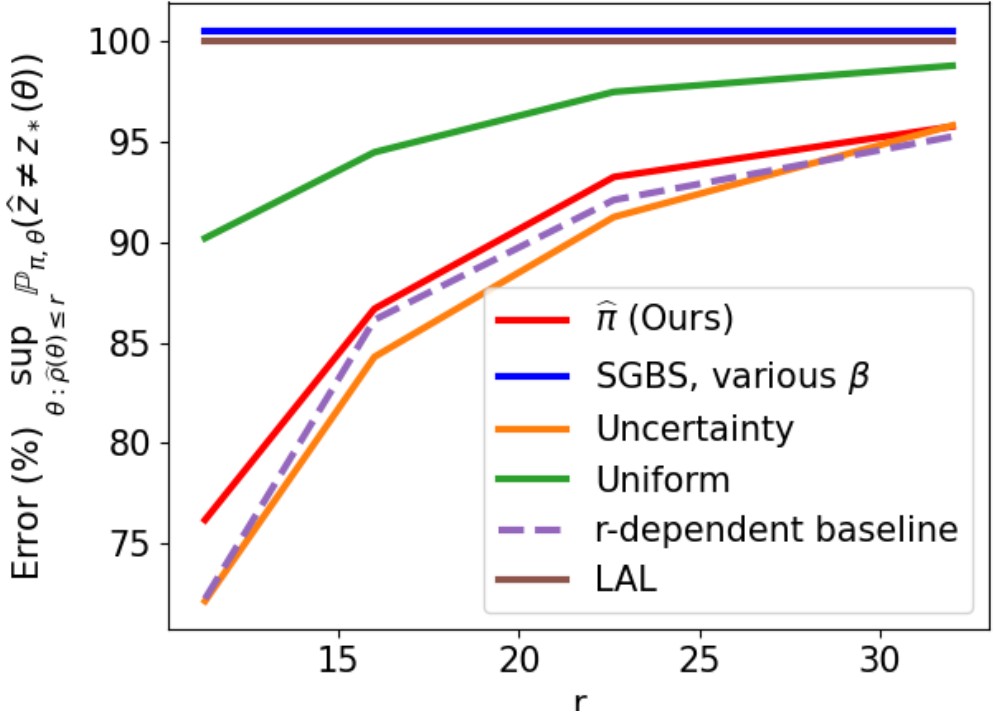

Figure 12: Full scale of Figure 6

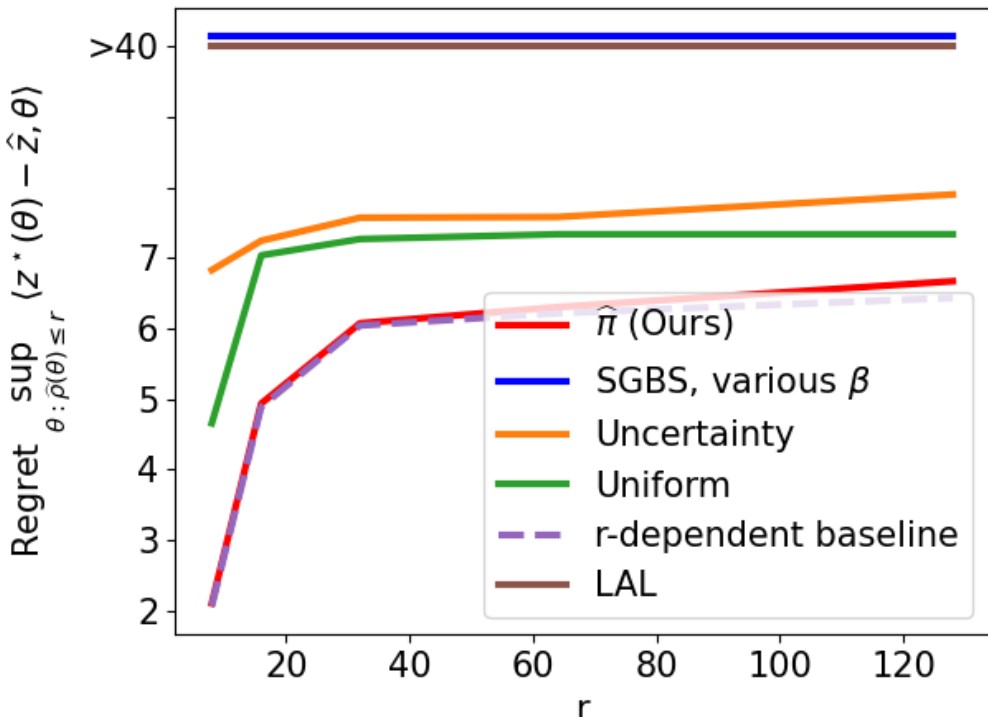

Figure 13: Full scale of Figure 7

## G  UNCERTAINTY SAMPLING

We define the symmetric difference of a set of binary vectors, $\text{SymDiff}(\{z_1, ..., z_n\}) = \{i : \exists j, k \in [n] \ s.t., z_j^{(i)} = 1 \wedge z_k^{(i)} = 0\}$, as the dimensions where inconsistencies exist.

---

**Algorithm 3:** Uncertainty sampling in very small budget setting

---

1  Input: $\mathcal{X}, \mathcal{Z}$
2  **for** $t = 1, ..., T$ **do**
3      $\widehat{\theta}_{t-1} = \text{argmin}_\theta \sum_{s=1}^{T} (y_s - \langle x_s, \theta \rangle)^2$
4      $\widehat{\mathcal{Z}} = \{z \in \mathcal{Z} : \max_{z' \in \mathcal{Z}} \langle z', \widehat{\theta}_{t-1} \rangle = \langle z, \widehat{\theta}_{t-1} \rangle\}$
5      **if** $|\widehat{\mathcal{Z}}| = 1$ **then**
6          $\widehat{\mathcal{Z}}_t = \widehat{\mathcal{Z}} \bigcup \{z \in \mathcal{Z} : \max_{z' \in (\mathcal{Z} \backslash \widehat{\mathcal{Z}})} \langle z', \widehat{\theta}_{t-1} \rangle = \langle z, \widehat{\theta}_{t-1} \rangle\}$
7      **else**
8          $\widehat{\mathcal{Z}}_t = \widehat{\mathcal{Z}}$
9      **end**
10     Uniformly sample $I_t$ from $\text{SymDiff}(\widehat{\mathcal{Z}}_t)$
11     Pull $x_{I_t}$ and observe $y_t$
12 **end**

---

## H  LEARNING TO ACTIVELY LEARN ALGORITHM

To train a policy under the learning to actively learn setting, we aim to solve for the objective

$$\min_\psi \mathbb{E}_{\theta \sim \widehat{\mathcal{P}}}[\ell(\pi^\psi, \theta)]$$

where our policy and states are parameterized the same way as Appendix C for a fair comparison. To optimize for the parameter, we take gradient steps like (8) but with the new sampling and rollout where $\widetilde{\theta}_i \sim \widehat{\mathcal{P}}$. This gradient step follows from both the classical policy gradient algorithm in reinforcement learning as well as from recent LAL work by Kveton et al. (2020).

Moreover, note that the optimal policy for the objective must be deterministic as justified by deterministic policies being optimal for MDPs. Therefore, it is clear that, under our experiment setting, the deterministic LAL policy will perform poorly in the adversarial setting (for the same reason why SGBS performs poorly).

## I  20 QUESTIONS SETUP

Hu et al. (2018) collected a dataset of 1000 celebrities and 500 possible questions to ask about each celebrity. We chose 100 questions out of the 500 by first constructing $\bar{p}'$, $\mathcal{X}'$ and $\mathcal{Z}'$ for the 500 dimensions data, and sampling without replacement 100 of the 500 dimensions from a distribution derived from a static allocation. We down-sampled the number of questions so our training can run with sufficient $M$ and $L$ to de-noise the gradients while being prototyped with a single GPU.

Specifically, the dataset from Hu et al. (2018) consists of probabilities of people answering *Yes / No / Unknown* to each celebrity-question pair collected from some population. To better fit the linear bandit scenario, we re-normalize the probability of getting *Yes / No*, conditioning on the event that these people did not answer *Unknown*. The probability of answering *Yes* to all 500 questions for each celebrity then constitutes vectors $\bar{p}'^{(1)}, ..., \bar{p}'^{(1000)} \in \mathbb{R}^{500}$, where each dimension of a give $\bar{p}_i'^{(j)}$ represents the probability of yes to the $i$th question about the $j$th person. The action set $\mathcal{X}'$ is then constructed as $\mathcal{X}' = \{\mathbf{e}_i : i \in [500]\}$, while $\mathcal{Z}' = \{z^{(j)} : [z_i^{(j)}] = \mathbf{1}\{\bar{p}_i^{(j)} > 1/2\}\} \subset \{0, 1\}^{1000}$ are binary vectors taking the majority votes.

To sub-sample 100 questions from the 500, we could have uniformly at random selected the questions, but many of these questions are not very discriminative. Thus, we chose a "good" set of queries based on the design recommended by $\rho_\star$ of Fiez et al. (2019). If questions were being answered noiselessly in response to a particular $z \in \mathcal{Z}'$, then equivalently we have that for this setting

$\theta = 2z - 1$. Since $\rho_\star$ optimizes allocations $\lambda$ over $\mathcal{X}'$ that would reduce the number of required queries as much as possible (according to the information theoretic bound of (Fiez et al., 2019)) if we want to find a single allocation for all $z' \in \mathcal{Z}$ simultaneously, we can perform the optimization problem

$$\min_{\lambda \in \Delta^{(|X|-1)}} \max_{z' \in \mathcal{Z}'} \max_{z \neq z'} \frac{\|z' - z\|^2_{(\sum_i \lambda_i x_i x_i^T)^{-1}}}{((z' - z)^T (2z' - 1))^2}.$$

We then sample elements from $\mathcal{X}'$ according to this optimal $\lambda$ without replacement and add them to $\mathcal{X}$ until $|\mathcal{X}| = 100$.

## J  JESTER JOKE RECOMMENDATION SETUP

We consider the Jester jokes dataset of Goldberg et al. (2001) that contains jokes ranging from pun-based jokes to grossly offensive. We filter the dataset to only contain users that rated all 100 jokes, resulting in 14116 users. A rating of each joke was provided on a $[-10, 10]$ scale which was shrunk to $[-1, 1]$. Denote this set of ratings as $\hat{\Theta} = \{\theta_i : i \in [14116], \theta_i \in [-1, 1]^{100}\}$, where $\theta_i$ encodes the ratings of all 100 jokes by user $i$. To construct the set of arms $\mathcal{Z}$, we then clustered the ratings of these users to 10 groups to obtain $\mathcal{Z} = \{z_i : i \in [10], z_i \in \{0, 1\}^{100}\}$ by minimizing the following metric:

$$\min_{\mathcal{Z}:|\mathcal{Z}|=10} \sum_{i=1}^{14116} \max_{z_* \in \{0,1\}^{100}} \langle z_*, \theta_i \rangle - \max_{z \in \mathcal{Z}} \langle z, \theta_i \rangle.$$

To solve for $\mathcal{Z}$, we adapt the $k - means$ algorithm, with the metric above instead of the $L - 2$ metric used traditionally.

