# OpenReview forum: "Learning to Actively Learn: A Robust Approach"
_ICLR.cc/2021/Conference — Reject_

### Official Review · AnonReviewer2 · 2020-10-25
**The authors introduce a novel, robust approach to Learning to Actively Learn. Even though this is an interesting solution to an important problem, the paper is difficult to follow and the evaluation results needs significant improvements.**

**Rating:** 5
**Confidence:** 2

**Review:**

Even though the paper introduces an interesting solution to an important problem, it suffers from two main weaknesses:
- it lacks an lacks an intuitive explanation of the proposed approach, thus making it hard to read
- the Empirical Evaluation seems to lack both a unifying theme and certain critical details

In order to make the paper easier to read, it would be great to add, right after the introduction, a section that explains in an intuitive manner the proposed approach. The authors should choose a motivating example  (ideally, it should be a real world domain, but it could also a be a simplified, synthetic one)  on which to explain all the basics of the proposed approach: which are the equivalence classes, how is the adversarial training leveraged, and how/when will one suffer the catastrophic loss referred to at the very bottom of page 1.

The empirical validation will greatly benefit by tightening the various arguments.

For 20 Questions:
- the scalability issue should not be confined to APPENDIX I: what would have taken to train on the entire dataset? What would take to use a dataset with 10x celebrities, 10x questions, or both?
- the statement "Uncertainty sampling even outperforms our r-dependent baseline by a bit which in theory should not occur – we conjecture this is due to insufficient convergence of our policies or local minima" certainly deserves a paragraph of its own, and a lot deeper an explanation than currently provided
- the accuracy results in Table 1 deserve a discussion:  why is your approach better than SGBS, when in 5 out of 6 settings SGBS has farr better results than your method?
- last but not least, you define should explain how "Accuracy" is computed (Table 1 is the only place in the paper where this term is used), and you should also use the "Win Rate" metric so that we can have an apples-to-apples comparison with the results in the original paper

For Joke Jester:
- please explain why your are restricting the dataset to include only jokes that are rated by all users. Is it a scalability issue? Or something else?
- you should also provide results in terms of Normalized Mean Absolute Error (NMAE), so that we can compare with results in the original paper
- In Table 2, the Uncertainty Sampling approach has better performance than your proposed method, even though in Figure 7 it seems to be doing consistently worse. Could you please discuss this phenomenon?

Other comments:
- it is very difficult to make sense of Figures 2 & 3 even when zooming-in of a large display.
- In Figure 2, for r > 20, there seems to always be 2-3 policies better than the proposed one; however, the image is so "crowded" in the r <= 20 region that it is impossible to see what is going on even at max-magnification. The authors should explain in prose what is going on & why they consider their policy best-overall. They should also provide in an APPENDIX several 1:1 graphs that compare the proposed policy against each of the other main contenders.
- In the paragraph that covers Figure 3, the authors write "π∗ performs almost as well as the r-dependent baseline policies over the range of r." However, Figure 3 does not include the r-dependent baseline.
- the authors should be consistent in notation: Figure 4 uses the LAL acronym, while Figure 5 uses the full name for it.

---

> ### Author Response · Authors · 2020-11-14
> **Response to AnonReviewer2**
>
> Thank you for your helpful comments. Please take a look at our general response first as we address some of the common comments.
>
> To address your specific concerns:
> 1. 20 Questions:
>
> a. We agree with the reviewer that r-dependent baseline performing worse than uncertainty sampling should not be expected theoretically. However, the differences between the baseline and uncertainty sampling are rather minimal. We do believe this is due to either insufficient training time or a local minima in our non-convex optimization. To be sure requires further investigation which we will pursue as soon as possible.
>
> b. We cannot create an apple-to-apple comparison since the original paper was trained under a different setting. In particular, they assume the probability scores theta_i to be known during training and build it as part of their state representation. Moreover, they trained and evaluated their policy on the same 1000 theta’s. As discussed in our response to common comments, this is not a reasonable setting in practice. In fact, although their probabilities are collected from thousands of users, we have confirmed with the original authors that for some question-celebrity pairs, only a few users (less than 5) have labelled the pairs. The theta’s are therefore subject to strong uncertainty when evaluating on a different population.
>
> 2. Joke Jester
>
> a. The authors agree that comparing our active policies with the passive algorithm proposed in the original paper could be valuable. We measured our performance based on a standard metric for multi-armed bandits, but we agree that NMAE metric is also a good metric for this task. We don’t foresee any difficulty generalizing our framework to the NMAE metric, and we will experiment with this as soon as possible after resolving the concerns in 1.a.
>
> b. During evaluation, there isn't any scalability issue since we are no longer doing policy optimization--just rolling out the policy which is just no harder than evaluating a neural network each step. We are only evaluating the fully rated user responses since our metric, simple regret, is defined based on knowing all elements of theta at test time.
>
> 3. In Figure 2 & 3, each of pi_k is optimized to be robust only in their corresponding equivalence class (see Figure 1 as a demonstration). As a result, each pi_k should be the optimal policy for its equivalence class of problems, but may not generalize well to other classes. Therefore, we formulate, for k = 1...K, pi_k’s performances as the r-dependent baseline, which may be unachievable by any single policy.  We define pi_* to minimize the additive performance gap between itself and the r-dependent baseline. Therefore, it generalizes well to all equivalence classes as demonstrated in Figure 2&3. In Figure 3, we are subtracting each policy’s performance by the r-dependent baseline. Therefore, the r-dependent baseline is inherently encoded as Gap(%) = 0. We’ve also updated the legend to “LAL”. The full-scaled figures have also been added to the Appendix.

---

### Official Review · AnonReviewer4 · 2020-10-28
**Not fit for publication in its current form**

**Rating:** 3
**Confidence:** 4

**Review:**

The paper sets out to address a class of pure exploration problems. Of particular interest to the authors seem to be settings where the sample complexity requirements are very stringent. A good example of such a task would be that of tuning the hyperparameters of an expensive training algorithm like BERT pre-training or ResNet.

Even though the problem domain is very interesting, the paper is just not fit for publication in this form for the following reasons:

1- The write-up is just horrific:

  a- The problem setting doesn't mention anything about intermediate rewards, which was very confusing because the reader is given the impression that learning happens based on just the total reward at the end of the trajectory.

  b- There are some definitions that just don't make sense: for instance, Equation (1) is subtracting a number from a policy. My guess is that the equation is missing some parentheses, but what's the point of formulas if they're not precise. Also, a more minor point is that the is no such thing as arginf: the whole point of inf is to deal with situations where the function doesn't have a minimum, e.g. inf_{x>0} 1/x = 0. What you mean is argmin, in which case you need some sort of compactness assumption in the policy space.

  c- It's completely unclear what the point of Prop 1 is.

  d- More generally, the paper could really benefit from a table of notations so the reader doesn't have to keep jumping back and forth.

2- It's really unclear what the contributions of the paper are: there are tons of papers on linear bandits in the pure exploration setting, so what does this paper add? There is no sample complexity analysis in there, which would be fine if the paper was a solid experimental paper, which it is not as discussed in the next point.

3- I find the experiments very unconvincing: as mentioned above, a really good motivating use-case is hyperparameter tuning for a very complex models or at least a diverse set of examples. A good example of a paper that does a satisfactory job of providing convincing experimental results is the Hyperband paper [Li et al, 2017], which render the paper valuable despite its weak theoretical results.

Please get one of your colleagues to read the paper before resubmitting it.

---

> ### Author Response · Authors · 2020-11-13
> **Response to AnonReviewer4**
>
> Thank you for your time in providing a review. However, as your colleague and a member of this community that takes pride in training students to become researchers, I plead with you to maintain a respectful and decent tone within your reviews. Denunciations like “horrific” are not appropriate in any review. Let us address some of your comments. While the pure exploration linear bandits has a rich literature of successful algorithms, they are only applicable in the large sample regime, e.g., a measurement budget in the hundreds. As you note in your review, we are interested in the very stringent budget regime, say T=20 questions, where one has little hope of proving any non-trivial results. The contribution of our proposed framework is a more theoretically justified approach to this very difficult regime in which only heuristics exist. We present three experimental setups (synthetic, 20-questions, Jester) under two evaluation schemes (instance-dependent worst-case, average case) (also see comments above) that we believe provide convincing evidence of performance.

---

### Official Review · AnonReviewer3 · 2020-10-28
**Interesting paper. But I have some concern.**

**Rating:** 4
**Confidence:** 3

**Review:**

Summary:
In this paper, the authors present an adaptive model that can learn a good policy by adversarial training. The proposed model is based on theoretic lower bounds. They focus on the setting when the query budget is small and conduct some experiments to verify the proposed method.

Pros:
- The authors derive some theoretical results.
- Using adversarial training is interesting.

Cons:
- The notations are too complicated and it's hard to follow the author's thought.
- I feel like the query budget in the experiment is very small (T=20). This setting might be unrealistic. I would like to see if the proposed method can have good performance when the budget is reasonably large.
- The current experimental results are not very convincing for me. Uncertainty or SGBS seems to be a good choice in some cases as well. Maybe the author can do experiments on more datasets and see if the proposed method performs better in most datasets.
- I notice that the authors subsample the training examples. It seems that the training the proposed method is slow so the number of training data and the budget are limited.

Typos:
- I thought in Figure 2, 3, 4, 6, 7, \hat{pi} should be pi_*

---

> ### Author Response · Authors · 2020-11-13
> **Response to AnonReviewer3**
>
> Thank you for your helpful comments. Please see our general response to the common comments as we try to address some of your concerns about this work.

---

> > ### Comment · AnonReviewer3 · 2020-11-25
> > **Current results are still not convincing for me**
> >
> > Thanks for your reply.  But current results are still not convincing for me.
> > the reasons are as follows:
> > - Intuitively, the proposed should learn a competitive policy to the best method, but if we focus on the accuracy, it's not the case. Maybe consider more datasets can be helpful. If conducting experiments on 8 or more datasets and the proposed method is usually the top2 best, then I would be convinced.
> > - I would like to see if the proposed method can have good performance when the budget is reasonably large. Even negative results are interesting since we can use the proposed methods at the beginning and use other algorithms when we have enough labeled examples.
> > - The authors may have to consider some approaches working on cold-start or active learning experience transfer work. Since their methods can handle the small budget as well. Especially in [2], their results seem to work on small budget as well.
> >
> > Some references:
> >
> > [1] Can active learning experience be transferred? In Proceedings of the IEEE International Conference on Data Mining, pages 841--846, 2016.
> >
> > [2] Learning Active Learning from Data. Advances in Neural Information Processing Systems, 2017

---

> > > ### Author Response · Authors · 2020-11-25
> > > **Further clarification on the goals of the paper**
> > >
> > > Thank you for your reply.
> > >
> > > In this paper we try to address the weak-in-robustness issue for existing algorithms under a small budget setting. As we have argued in our rebuttal, such setting is common in real applications. Also, in tasks related to medical recommendation for example, one would be willing to trade average-case accuracy for the robustness of the algorithm. Indeed, such instance dependent worse-case accuracy guarantees are the standard for multi-armed bandits in the frequentist setting.
> > >
> > > *As a secondary argument in this paper, we try to argue that given the robustness of our policies, the average-case accuracy is still on par with the other algorithms optimized for the average-case accuracy. However, first and foremost, our primary contribution is to propose a training framework for policies to perform well on the instance-dependent worst-case metric.* It seems like the reviewer disagree with the claim that our policies are comparable under the average-case accuracy. In this case, we are happy to play down this particular argument and note the real tradeoff between instance-dependent worst-case accuracy and average-case accuracy in future versions.
> > >
> > > To put the tradeoff more concretely, as discussed in the related work about the second paper mentioned by the reviewer, pure learning to actively learn algorithms are optimized with respect to the average-case performance over a fixed set of problem instances which can be considered as fixed, explicit prior distribution. Let’s call it P_theta. On the other hand, our objective is more robust in two ways: 1) we consider the entire space of problem instances; 2) given a problem instance, we guarantee the worst possible outcome of the algorithm. Therefore, in order to optimize for an instance-dependent worst-case objective, one would necessarily need to tradeoff the optimal average-case performance for 1) thetas in the support of P_theta that have relatively poor accuracy; 2) theta's that are outside the support of P_theta, which one may encounter at test time in the real world.
> > >
> > > Taking the second paper for example, when put into our experiments, it will perform poorly under the instance-dependent worst-case metric. In fact, any policy optimized for the average case accuracy is provably guaranteed to fail catastrophically in the small budget setting. This is because all such policies are deterministic, meaning given a history, with probability 1, the policy will choose to pull one of the arms/make a specific query. It is easy to prove that deterministic algorithms under the small budget setting could be easily tricked by an adversary and we can provide a short proof if the reviewer find it necessary. We should also note that policies such as uncertainty sampling and our polices do not pull an arm deterministically given histories.

---

### Official Review · AnonReviewer1 · 2020-10-29
**Differentiable instance dependent learning**

**Rating:** 7
**Confidence:** 4

**Review:**

The paper "Learning to Actively Learn" proposes a differentiable procedure to design algorithms for adaptive data collection tasks. The framework is based on the idea of making use of a measure of problem convexity (each problem is parametrized by a parameter theta) to solve solving a min-max objective over policies. The rationale behind this objective is that the resulting policy of solving this min-max objective should be robust to the problem complexity. The algorithm then proceeds to sample problem instances and making use of a differentiable objective to find a policy parametrized by a parameter psi, which could be parametrizing a neural network.

One potential drawback is that the authors assume the dynamics of the problem instance rewards are known by the learner (for example they are a gaussian), which is necessary for computing policy gradients through the policy parametrization. A second drawback lies in the problem tessellation over the theta space. As it is written the method does not seem to scale beyond very small dimensional problem instances, since otherwise the value N would have to be exponential in the dimension, and therefore intractably large. The paper falls within the differentiable "meta-learning" for bandits literature, and it does a good job of placing itself within that literature. It also has a convincing experimental section. Other works in the area have not tackled the problem that the authors set themselves to solve: designing algorithms that can adaptively perform well depending on the instance they are fed.

I also find particularly interesting the use of the model complexity balls that can be defined using other existing results in the literature such as in the case of transductive linear bandits. I would suggest to add more explanation to what r is earlier in the paper as it is hard on the reader after a first read. Overall I think this is nice work. The paper itself is more applied than theoretical but I think it is appropriate for ICLR.

---

> ### Author Response · Authors · 2020-11-13
> **Response to AnonReviewer1**
>
> Thank you for your helpful comments. Please see our general response to the common comments as we try to address some of your concerns about this work.

---

### Author Response · Authors · 2020-11-13
**General Response to Common Comments by the Reviewers**

We thank all the reviewers for their time and helpful reviews.

We have significantly improved the *readability* of our paper. In particular, we
1. added a clear main objective of the paper
2. added a section explaining the complexity function and equivalence classes
3. added more details of different notations used
4. added a paragraph detailing the differences between the two metrics we evaluate our policies on (instance-dependent worst-case and average case).

Due to some confusion over the experiments, we wish to clarify the difference between the *instance-dependent worst-case versus average case* criteria. The instance-dependent criteria mirrors the theoretical guarantees of frequentist algorithms for multi-armed bandits--any instance classified as a certain difficulty should achieve a certain probability of success under a fixed budget. Our algorithm is trained specifically for the instance-dependent criteria and fairs far better than other algorithms that are not. For example, SGBS fails catastrophically (see Figures 4,6 and 7). We also evaluate on an “average case” criteria derived from a benign distribution over thetas, averaged over the observed thetas present in the datasets when possible. We could easily train a policy to perform optimally with respect to the distribution and indeed this is exactly what previous works do (see refs in learning to actively learn related work and original 20Q paper by Hu et al. (2018)). We explicitly avoid this because we find it to be unrealistic: choosing the distribution over thetas is extremely difficult until you’ve already seen data to train on, and if the prior is incorrectly specified, performance could suffer significantly. . Our approach is trained for the instance dependent setting to avoid defining a prior in an attempt to increase robustness, but this means we may not perform optimally in both settings. We could obviously perform dataset selection to ensure we always win in both settings decisively, but the intention of our experiments is to demonstrate that the tradeoffs necessarily exist.

Our paper targets the *very small measurement budget* regime where only tens of measurements (instead of hundreds or thousands) are possible. This is very frequently the setting encountered in e-commerce recommendation settings where order histories are small. And indeed, our experiments on 20 Questions and Jester Joke recommendation also fall into this regime.  While there are many theoretical bandit papers with sharp sample complexity results, they only apply when the measurement budget is very large since they tend to appeal to loose concentration inequalities. Our proposed framework is directly motivated by the fact that these existing results are not only vacuous in this small budget regime, but also that they tend to perform very poorly in practice in this regime, acting no differently than random sampling.

In response to concerns of *computational complexity* we acknowledge that policy gradient with adversarial training is a demanding procedure. We subsampled our datasets in cases to have all known responses and to have our models fit in the memory of our GPU with a large batch size. Utilizing multiple GPUs would allow our methods to scale to larger problems. Reviewer 1 points out that if the N particles are interpreted as a cover of Theta-space, N would have to be exponential in the dimension. However, due to the fact that each particle is constantly performing gradient descent to adapt to the policy pi, we believe N need only be large enough to represent the set of worst-case alternatives with respect to pi which may have significantly better dependence on the dimension.

---

### Decision · Program_Chairs · 2021-01-07
**Final Decision**

**Decision:**

Reject

**Comment:**

The authors present an adaptive model that learns a good policy by adversarial training, focusing on the setting where the query budget is very small. Some experiments are carried out to validate the proposed method. The reviewers' opinions turned out to be split on this paper. On one hand, all reviewers appreciated the idea of the problem and recognized its importance. On the other hand, there are have been multiple concerns regarding readability (but that has improved during the discussion) and about the empirical validation/evaluation.
Based on the above, as well as my own reading, I believe this paper contains interesting ideas but, as it currently stands, is not ready for publication.